# Acrylic acid and related dimethylated sulfur compounds in the Bohai and Yellow Seas during summer and winter

Xi Wu[2,3], Pei-Feng Li[2,3], Hong-Hai Zhang[1,2,3], Mao-Xu Zhu[1,2,3], Chun-Ying Liu[1,2,3], Gui-Peng Yang[1,2,3]

[1]Frontiers Science Center for Deep Ocean Multispheres and Earth System, and key Laboratory of Marine Chemistry Theory and Technology, Ministry of Education, Qingdao, 266100, China

[2]Laboratory for Marine Ecology and Environmental Science, Qingdao National Laboratory for Marine Science and Technology, Qingdao, 266071, China

[3]College of Chemistry and Chemical Engineering, Ocean University of China, Qingdao, 266100, China

*Correspondence to*: Mao-Xu Zhu (zhumaoxu@ouc.edu.cn); Chun-Ying Liu (roseliu@ouc.edu.cn)

**Abstract.** Spatio-temporal distributions of dissolved acrylic acid (AAd) and related biogenic sulfur compounds including dimethylsulfide (DMS) and dissolved and total dimethylsulfoniopropionate (DMSPd and DMSPt) were investigated in the Bohai Sea (BS) and Yellow Sea (YS) during summer and winter. AAd and DMS production from DMSPd degradation and AAd degradation were analyzed. Significant seasonal variations of AAd and DMS(P) were observed. AAd exhibited similar distributions during summer and winter, i.e., relatively high values of AAd occurred in the BS and the northern YS, and the concentrations decreased from inshore to offshore areas in the southern YS. Due to strong biological production from DMSP and abundant terrestrial inputs from rivers in summer, the AAd concentrations in the surface seawater during summer (30.01 nmol L$^{-1}$) were significantly higher than those during winter (14.98 nmol L$^{-1}$). The average concentration sequence along the transects during summer (AAd > DMSPt > DMS > DMSPd) showed that particulate DMSP (DMSPp) acted as a DMS producer, and terrestrial sources of AAd were present; in contrast, the sequence in winter was AAd > DMSPt > DMSPd > DMS. High values of AAd and DMS(P) were mostly observed in the upper layers, with occasional high values at the bottom. High AAd concentrations in the porewater, which could be transported to the bottom water, might result from the cleavage of intracellular DMSP and reduce bacterial metabolism in sediments. In addition, the degradation/production rates of biogenic sulfur compounds were significantly higher in summer than in winter, and the removal of AAd was primarily attributed to microbial consumption. Other sources of AAd existed besides the production from DMSPd.

## 1 Introduction

Dimethylsulfide (DMS), which is biologically derived from the enzymatic cleavage of dimethylsulfoniopropionate (DMSP), is the dominant volatile sulfur compound released from the ocean to the atmosphere (Lovelock et al., 1972; Dacey and Wakeham, 1986). The annual emission of DMS from the ocean contributes 28.1 (17.6–34.4) Tg S to the atmosphere (Lana et al., 2011). Moreover, DMS is correlated with the natural acidity of rain (Nguyen et al., 1992). DMS produced in surface waters can chemically influence the marine system, global sulfur cycle, and global climate. The CLAW hypothesis proposes that the oxidation products of DMS are the major sources of cloud condensation nuclei (CCN), leading to an increase in aerosol albedo over the ocean and, consequently, to a decrease in solar radiation on the Earth's surface (Charlson et al., 1987; Malin et al., 1992; Zindler et al., 2012), although recent studies argued that other sources (e.g., bubble bursting at the ocean surface) are the major contributors to CCN on global scales (Quinn and Bates, 2011). Therefore, more studies are needed to further our understanding of the potential links between DMS and climate change.

DMSP, the biochemical precursor of DMS (Malin and Erst, 1997; Alcolombri et al., 2015), is produced by marine phytoplankton and marine heterotrophic bacteria (Keller et al., 1989; Curson et al., 2017). As an antioxidant, a cryoprotectant, and an osmolyte in marine phytoplankton, the production of DMSP is influenced by environmental parameters such as salinity (Stefels, 2000), temperature (Kirst et al., 1991), and oxidative stress (Sunda et al., 2002). DMSP distributions are also controlled by phytoplankton species, among which coccolithophorids, dinoflagellates, and prymnesiophytes are high-producing algae of DMSP (Keller et al., 1989), and diatoms, flagellates, Prochlorophytes and cyanobacteria are low producers of DMSP (McParland and Levine, 2019). Furthermore, DMSP provides considerable sulfur and carbon sources for the microbial food web. In addition, the degradation of DMSP occurs through two main pathways. The dominant pathway is demethylation, a complicated process generating different ultimate products through different enzymes possibly including methanethiol, hydrogen sulfide, and acrylic acid (AA) (Taylor and Visscher, 1996; Bentley and Chasteen, 2004; Reisch et al., 2011). The other pathway is enzymatic cleavage of DMSP into equimolar DMS and AA by phytoplankton (Steinke et al., 2002) and bacteria (Ledyard and Dacey, 1996); this is a minor pathway that contributes, on average, only 10% to DMSP degradation. (Reisch et al., 2011).

AA is chemically the simplest unsaturated carboxylic acid, and in coastal seawater, it is not only derived from DMSP cleavage but also from anthropogenic contamination via river discharges (Sicre et al., 1994). The removal of AA occurs mainly through two mechanisms, i.e., photochemical degradation (Bajt et al., 1997; Wu et al., 2015) and microbial degradation (Noordkamp et al., 2000). AA plays diverse roles in marine systems. For example, AA is an important carbon source for the microbial community (Noordkamp et al., 2000), while it also acts as an antibacterial agent (Sieburth, 1960; Slezak et al., 1994). Furthermore, the presence of AA functions as grazing-activated chemical defense and thus inhibits the predation of phytoplankton by microzooplankton (Wolfe et al., 1997).

Many aspects of DMS and DMSP have been well documented, including spatio-temporal distributions, degradation, sea-to-air fluxes, and particle size fractionation (Lana et al., 2011; Levine et al., 2012; Yang et al., 2014; Espinosa et al., 2015). Recently, the biogeochemistry of AA in the oceans and the roles of AA in the marine sulfur cycle and the microbial community have received increasing attention globally. Kinsey et al. (2016) explored the effects of iron limitation and UV radiation on Phaeocystis antarctica growth and AA concentrations. The concentrations, biological uptake, and respiration of dissolved AA (AAd) were investigated in the northern Gulf of Mexico (Tyssebotn et al., 2017). Tan et al. (2017) and Wu et al. (2017) reported the spatial distributions of AA in the Changjiang Estuary and the East China Sea. Liu et al. (2016) investigated the spatial and diurnal variations of AA in the Bohai Sea (BS) and Yellow Sea (YS) during autumn and measured the apparent production rates of AA through DMSP degradation by incubations. However, seasonal variations, the source and removal of AA, and the key factors controlling these processes remain unclear; thus, further studies are needed to obtain a better understanding of the biogeochemical cycle of sulfur in the oceans. In this study, we investigate the horizontal and vertical distributions of AAd and related dimethylated sulfur compounds in the BS and YS in different seasons (summer and winter) to determine if temperature, phytoplankton and bacteria species and abundance are the key factors controlling AA dynamics. In addition, for the first time, we collect AAd samples in the porewater of surface sediment during summer in the BS and YS. We also examine the degradation of dissolved DMSP (DMSPd) and AAd simultaneously through on-deck incubations during summer and winter to understand the production and consumption mechanisms of AA, DMS, and DMSP, to explore the influencing factors (i.e., the changes in the bacteria species and abundance) of microbial degradation, and to discover other potential sources of AA. This study is expected to provide insightful information on sulfur cycling regarding AA in the marginal seas.

## 2 Material and methods

### 2.1 Study area

The BS, the largest inner sea in China, is surrounded by Tianjin City, Hebei Province, and the Shandong and Liaodong Peninsulas. The total water area of the sea is $7.7 \times 10^4$ km$^2$ and the average water depth is 18 m. The hydrological conditions of the BS are substantially influenced by discharges from over 40 rivers, including the Yellow River, Haihe, Daliaohe, and Luanhe (Ning et al., 2010). Especially, the Yellow River, the world's second largest river in terms of sediment load, brings large amounts of particulates and nutrients to the BS. The YS, which is separated from the BS by

the Bohai Strait, is a shallow semi-enclosed marginal sea located between the Chinese mainland and the Korean Peninsula, with a total water area of $3.8 \times 10^5$ km$^2$ and a mean depth of 44 m. The YS is divided into the northern Yellow Sea (NYS) and the southern Yellow Sea (SYS) by a line between Chengshan Cape on the Shandong Peninsula and Changshanchuan on the Korean Peninsula. The BS and YS are substantially affected by complicated water currents and two main water masses including the Bohai Sea Coastal Current (BSCC), the Yellow Sea Coastal Current (YSCC), the Korea Coastal

Current (KCC), the Yellow Sea Warm Current (YSWC), the Changjiang River Diluted Water (CRDW), and the Yellow Sea cold water mass (YSCWM) (Lee et al., 2000; Su, 1998) (Fig. 1). Moreover, anthropogenic pollution on both the China and Korea coasts has notable effects on the ecosystems including species diversity and community structure of phytoplankton and benthos in the BS and YS (Liu et al., 2011).

### 2.2 Sampling

Two cruises were conducted aboard the R/V ''Dong Fang Hong 2'' in the BS and YS from August 17th to September 5th 2015 (summer) and from January 14th to February 1st 2016 (winter). The summer cruise covered 52 grid stations and three transects and the winter cruise comprised 39 grid stations and two transects (Fig. 1). Seawater samples were collected using 12 L Niskin bottles mounted on a Seabird 911+ Conductivity-Temperature-Depth (CTD) sensor (Sea-Bird Electronics, Inc., USA). Temperature and salinity were measured by the CTD sensor. Water samples were transferred

from the Niskin bottles to 250 mL brown glass bottle through silicone tubing. While filling the bottles, the samples were allowed to overflow from the top of the bottle to eliminate any headspace to minimize partitioning into the gas phase. Sediments were collected using a stainless-steel box-corer and were sub-sampled to a depth of ca. 3 cm at 12 stations during summer cruise, as shown in Table 1.

### 2.3 Analytical procedures

The DMS concentrations of all samples were measured onboard immediately after sampling with a purge-and-trap technique modified from Andreae and Barnard (1983) and Kiene and Service (1991). A 2 mL aliquot of seawater sample was extracted from the 250 mL brown glass bottle using a 2 mL glass syringe and was filtered by syringe filtration through a 25 mm Whatman glass fiber (GF/F) filter (Li et al., 2016); the sample was directly injected into a glass bubbling chamber and extracted with high purity nitrogen at a flow rate of 40 mL min$^{-1}$ for 3 min. Then, the sulfur gases were dried using

Nafion gas sample dryer (Perma Pure, USA) and trapped in a loop of Teflon tubing immersed in liquid nitrogen (-196 °C). After extraction, the Teflon tubing was heated in boiling water, and the desorbed gases were introduced into a 14B gas chromatograph (Shimadzu, Japan) equipped with a flame photometric detector and a 3 m × 3 mm glass chromatographic column packed with 10% DEGS on Chromosorb W-AW-DMCS. The analytical precision of DMS was generally better than 10% and the detection limit was 0.4 nmol L$^{-1}$ (Yang et al., 2015a).

A 4 mL aliquot of seawater was filtered under gravity through a 47 mm Whatman GF/F filter (Kiene and Slezak, 2006) for DMSPd analysis. A 10 mL aliquot of seawater without filtering was used for total DMSP (DMSPt) analysis. For an

even DMSP concentration and the oxidation of endogenous DMS, 100 μL and 40 μL of 50 wt% sulfuric acid were added to the samples for DMSPt and DMSPd analysis, respectively (Shooter and Brimblecombe, 1989). The DMSPt and DMSPd samples were incubated in the dark at room temperature for 2 d to oxidize pre-existing gaseous DMS fully.

Before analysis, the samples were injected with 300 μL of 10 mol L$^{-1}$ KOH solutions and stored in the dark at 4 °C for at least 24 h to allow a complete conversion of DMSP into DMS. The measured DMS concentration was used to estimate the DMSP concentration, according to 1:1 stoichiometry (Dacey and Blough, 1987). This method provided detection limits for DMS of 0.05-0.5 nmol L$^{-1}$. Details on the concentrations of DMS and DMSP in surface seawater have been published in Master theses (Jin, 2016; Sun, 2017).

Seawater samples for AAd analyses were collected directly from the Niskin bottles and filtered under gravity through a pre-cleaned 0.2 μm AS 75 Polycap filter capsule (a nylon membrane with a glass microfiber pre-filter enclosed in a polypropylene housing; Whatman Corporation, USA) (Wu et al., 2015). The filtrate was transferred to a 40 mL glass vial with a Teflon™-lined cap and stored at 4 °C. Porewater samples for AAd analyses were extracted from surface sediments via Rhizon soil moisture samplers (0.1 μm porous polymer, Rhizosphere Research, Wageningen, the Netherlands) according to Seeberg-Elverfeldt et al. (2005). All porewater samples were stored at 4 °C and filtered through 0.22 μm polyethersulfone syringe filters (Membrana Corporation, Germany) before analysis. The AAd seawater and porewater samples were analyzed using a high-performance liquid chromatograph (L-2000, Hitachi Ltd., Japan), according to Gibson et al. (1996). An Agilent SB-Aq-C18 column and the eluent of 0.35% H$_3$PO$_4$ (pH = 2.0) at a flow rate of 0.5 mL min$^{-1}$ were used to separate the AAd. The column eluate was detected by a UV detector at 210 nm. The analytical precision was between 1.3% and 1.6%, and the detection limit was 4 nmol L$^{-1}$ (Liu et al., 2013).

For the chlorophyll *a* (Chl *a*) analysis, 300 mL of seawater was filtered through Whatman GF/F filters. Then the filtrates were soaked in 10 mL of 90% acetone and kept in the dark at 4 °C. The contents of Chl *a* were measured after 24 h using an F-4500 fluorescence spectrophotometer (Hitachi, Japan), according to Parsons et al. (1984). In addition, the nutrient concentrations (including PO$_4^{3-}$, NO$_3^-$, NO$_2^-$, NH$_4^+$, and SiO$_3^{2-}$) were analyzed using an automatic nutrient analyzer (Auto Analyzer 3, SEAL Analytical, USA). The phytoplankton data recored by Utermöhl method and bacteria data measured by qPCR were collected from Zhang (2018) and Liang et al. (2019), respectively. The analytical samples for DMS, DMSPd, DMSPt, AAd, Chl *a*, and the nutrients were run in duplicate.

## 2.4 Incubation experiments

The incubation experiments for DMSPd and AAd degradation were conducted on deck using seawater collected at stations H19, H26, B12, B17, B53, and B63 in summer and at H19, H26, B12, and B16 in winter according to Wu et al. (2017). We determined the degradation rates of DMSPd and the production rates of DMS and AAd by incubating unfiltered seawater samples in two 250 mL gas-tight glass syringes (wrapped in aluminum foil) in the dark at in situ temperatures. Before the incubations, 80 μL of concentrated DMSPd solution (0.2 mmol L$^{-1}$) was added to the two syringes to reach an initial concentration of DMSPd higher than 50 nmol L$^{-1}$. One syringe was used as the treatment group, and the other was used as the control by injecting it with glycine betaine (GBT, final concentration of 50 μmol L$^{-1}$, 1000× the concentration of added DMSPd). GBT inhibits microbial degradation of DMSP within a short time (Kiene and Service, 1993; Kiene and Gerard, 1995) because it is chemically and physiologically similar to DMSP and acts as a competitive inhibitor of DMSP (Kiene et al., 1998). After 0, 3, and 6 h, 25 mL aliquots of samples were taken from the incubations for measuring the DMSPd, DMS, and AAd concentrations. Linear regression equations were fit to the DMSPd, DMS, and AAd time course data, and the apparent rates were estimated as the differences between the slopes of the samples with and without GBT.

Two pathways of AAd degradation, i.e., photochemical consumption and microbial consumption, were experimentally investigated in this study. For the photochemical consumption of AAd, a drop of oversaturated $NaN_3$ solution was added to 300 mL seawater samples (the final concentration was approximately 1 mmol $L^{-1}$) to eliminate the microbial consumption of AAd. After filtration, the seawater samples were immediately injected into a 125 mL photic quartz tube and a 125 mL photophobic quartz tube (as a control) to initiate photochemical degradation; 10 mL aliquots of samples were taken for analyses of AAd at 0, 3, and 6 h. Linear regression equations were fit to the AAd time course data, and the photochemical degradation rates of AAd were calculated based on the differences between the slopes of the samples in the photic and photophobic quartz tubes (Wu et al., 2015).

For the microbial consumption of AAd, unfiltered seawater samples were used for incubations in 100 mL glass syringes (wrapped in aluminum foil) in the dark at in situ temperatures. Prior to incubation, concentrated AAd was added to one syringe to reach an initial concentration that was 10-50 times that of the background concentration. Another seawater sample without exogenous AAd addition was used as the control; 10 mL aliquots of samples were taken for determination of AAd at 0, 3, and 6 h. Linear regression equations were fit to the AAd time course data, and the microbial degradation rates of AAd were estimated as the differences between the slopes of the samples with exogenous AAd addition and the control (Wu et al., 2015). Duplicate samples were analyzed for AAd, DMS, and DMSPd in all the incubation experiments.

## 3 Results

### 3.1 Horizontal distributions of AAd in the BS and YS

In summer, the Chl $a$ contents in the surface seawater were in the range of 0.01-8.91 μg $L^{-1}$, with an average value of 1.95 ± 2.31 μg $L^{-1}$. The contents in the BS were relatively high, and an extremely high value (7.07 μg $L^{-1}$) occurred in the center of the sea, whereas the concentrations decreased gradually from the inshore to offshore areas in the NYS and the northern area of the SYS. The minimum value of Chl $a$ occurred in the center of the SYS, and the maximum was observed in the southern area of the SYS (station H37).

The AAd concentrations in the surface seawater during summer ranged from 10.53 to 92.29 nmol $L^{-1}$, with a mean of 30.01 ± 21.12 nmol $L^{-1}$, and the concentrations generally decreased from the north to the south (Fig. 2 and Table 1). The average values in the BS and the NYS were 40.76 ± 24.80 and 38.89 ± 22.61 nmol $L^{-1}$, respectively; these values were higher than the average value of the entire study area. In contrast, the mean value in the SYS was 18.02 ± 7.70 nmol $L^{-1}$, which was more than half of the average value of the entire study area, even though the Chl $a$ values were relatively high in the SYS. In addition, AAd was positively dependent on the temperature in the NYS (Table 2). Jin (2016) observed that DMS and DMSP showed decreasing trends from inshore to offshore areas (Fig. 3), which were coupled to the distribution pattern of Chl $a$. DMS and DMSP also exhibited higher values in the BS than in the YS, similar to the case of AAd.

In winter, the Chl $a$ contents in the surface seawater ranged from 0.16 to 0.99 μg $L^{-1}$ (mean: 0.47 ± 0.21 μg $L^{-1}$) and generally decreased from the inshore to offshore areas. The AAd concentrations ranged from 4.28 to 42.05 nmol $L^{-1}$ (mean: 14.98 ± 7.72 nmol $L^{-1}$), and high concentrations occurred near the Chengshan Cape where high values of Chl $a$, DMS, and DMSP, as well as high phytoplankton abundance were also observed (Figs. 2 and 3) (Sun, 2017; Zhang, 2018). Chl $a$, AAd, DMS, and DMSPd all showed declining trends from the inshore to offshore areas in the SYS. Note that the AAd concentrations in the BS (15.94 ± 10.49 nmol $L^{-1}$), the NYS (14.53 ± 7.64 nmol $L^{-1}$), and the SYS (14.91 ± 6.31 nmol $L^{-1}$) were not significantly different.

**3.2 Vertical distributions of AAd, DMS, and DMSP in the BS and YS**

In summer, the three transects B57-63, B12-17, and H19-26, which were located in the BS, the NYS, and the SYS, respectively, were chosen to investigate the vertical distributions of AAd, DMS, and DMSP. Along transect B57-63, the Chl $a$, AAd, DMS, DMSPd, and DMSPt concentrations were in the ranges of 0.15-7.07 µg L$^{-1}$ (mean 1.58 ± 1.88 µg L$^{-1}$), 11.08-73.06 nmol L$^{-1}$ (mean 36.36 ± 23.57 nmol L$^{-1}$), 2.57-8.79 nmol L$^{-1}$ (mean 5.51 ± 2.01 nmol L$^{-1}$), 0.72-3.37 nmol L$^{-1}$ (mean 1.56 ± 0.84 nmol L$^{-1}$), and 4.12-56.61 nmol L$^{-1}$ (mean 22.94 ± 21.28 nmol L$^{-1}$), respectively. All of the

compounds had high values in the upper layers. Meanwhile, Chl $a$ and AAd showed relatively high values at the bottom of station B61 and B57, respectively (Fig. 4).

Along transect B12-17, the Chl $a$ and DMS concentrations ranged from 0.18 to 2.87 µg L$^{-1}$ and from 0.74 to 15.76 nmol L$^{-1}$, with means of 0.92 ± 0.96 µg L$^{-1}$ and 7.37 ± 4.50 nmol L$^{-1}$, respectively. Low values of Chl $a$ occurred in the bottom seawater of the transect and in the water column of station B15, whereas Chl $a$ and DMS exhibited maximum values at

15 m depth at stations B13 and 25 m depth at station B15, respectively (Fig. 4). The concentrations of DMSPd, DMSPt, and AAd were in the ranges of 0.36-2.01 nmol L$^{-1}$, 1.90-63.03 nmol L$^{-1}$, and 12.77-102.988 nmol L$^{-1}$, with averages of 1.12 ± 0.48 nmol L$^{-1}$, 15.45 ± 17.98 nmol L$^{-1}$, and 34.60 ± 26.00 nmol L$^{-1}$, respectively. The concentrations generally declined with depth, and the highest concentrations were observed in the surface layers of stations B12 and B13. Yang et al. (2015a) also found maximum values of DMS and DMSP in the upper water column along transect B12-17 during late

fall, which were restricted mostly to the euphotic layer. High values of AAd also occurred in the bottom water of stations B13 and B17. DMSPd and DMSPt showed a strong positive correlation (Table 2), whereas AAd was not correlated with DMSP. The average value of AAd was more than 2 times that of DMSPt, the precursor of AAd, which demonstrated that terrestrial inputs contributed substantially to AAd along transect B12-17.

Transect H19-26 was affected by the YSCWM in summer, as indicated by low temperatures (<10 °C) below 40 m water

depth. A tidal front divided the transect into a well-mixed shallow water area (station H19) and a stratified deep-water area occupied by the YSCWM (stations H21-H26) (Fig. 4). The concentrations of Chl $a$, DMS, DMSPd, DMSPt, and AAd were in the ranges of 0.12-1.50 µg L$^{-1}$ (mean 0.58 ± 0.39 µg L$^{-1}$), 0.79-21.98 nmol L$^{-1}$ (mean 6.44 ± 5.14 nmol L$^{-1}$), 0.61-21.59 nmol L$^{-1}$ (mean 3.05 ± 4.92 nmol L$^{-1}$), 1.11-55.14 nmol L$^{-1}$ (mean 13.67 ± 12.90 nmol L$^{-1}$), and 13.19-85.86 nmol L$^{-1}$ (mean 22.24 ± 18.25 nmol L$^{-1}$), respectively. DMSPd, DMSPt, and AAd showed stratified distributions similar

to those of the temperature, whereas Chl $a$ and DMS did not. The Chl $a$ contents generally decreased from the inshore to offshore areas, with minimum values in the medium and bottom layers of the offshore stations. High values of sulfur compounds in the surface seawater and higher concentrations in the YSCWM region than in the well-mixed shallow water region were in agreement with the results of Yang et al. (2015b). In addition, there was a relatively high value of DMS in the bottom layer of station H23. There were no significant correlations between AAd, DMS, DMSPd, and DMSPt,

although these compounds showed similar patterns of spatial distribution. DMSPt had a positive correlation with temperature and a negative correlation with salinity (Table 2). Many other investigations also reported analogous correlations (Shenoy and Patil, 2003; Deschaseaux et al., 2014; Wu et al., 2017).

In winter, transect B57-63 was inaccessible to sampling due to frozen conditions; thus, we only report the results of transect B12-16 in the NYS and transect H19-26 in the SYS. Along transect B12-16, the Chl $a$, DMS, DMSPd, DMSPt,

and AAd concentrations were in the ranges of 0.17-1.56 µg L$^{-1}$, 1.12-4.56 nmol L$^{-1}$, 1.54-4.55 nmol L$^{-1}$, 5.33-24.50 nmol L$^{-1}$, and 13.94-27.69 nmol L$^{-1}$, with averages of 0.53 ± 0.43 µg L$^{-1}$, 1.99 ± 1.02 nmol L$^{-1}$, 2.92 ± 0.82 nmol L$^{-1}$, 11.44 ± 5.89 nmol L$^{-1}$, and 17.68 ± 5.21 nmol L$^{-1}$, respectively. Furthermore, Chl $a$, DMS, and DMSPt showed homogeneous distributions from the surface to the bottom, whereas DMSPd and AAd were heterogeneously distributed, with minimum values at the surface and maximum values at the bottom (Fig. 5).

Along transect H19-26, The concentrations of Chl $a$ and DMSPt ranged from 0.13 to 0.42 μg L$^{-1}$ and from 6.12 to 19.92 nmol L$^{-1}$, with means of $0.28 \pm 0.09$ μg L$^{-1}$ and $11.88 \pm 3.97$ nmol L$^{-1}$, respectively. The concentrations declined from the inshore to offshore areas, whereas DMS (0.52-1.35 nmol L$^{-1}$, average $0.96 \pm 0.29$ nmol L$^{-1}$) and DMSPd (1.92-6.06 nmol L$^{-1}$, average $3.06 \pm 1.07$ nmol L$^{-1}$) showed decreasing trends from the surface to the bottom (Fig. 5). The AAd concentrations ranged from 11.04 to 39.47 nmol L$^{-1}$ (mean $17.08 \pm 6.72$ nmol L$^{-1}$), and there were no significant differences along the transect H19-26, except for the maximum value at the bottom of station H24.

Along the three transects, high values of AAd, DMS, and DMSP occurred in the bottom water occasionally during summer and winter, which might have resulted from the release from porewater (Andreae, 1985) (Figs. 4 and 5). DMSP showed positive correlations with temperature and negative correlations with salinity along the three transects during summer, whereas DMS and DMSP had negative correlations with temperature and salinity during winter; these results may be attributed to the co-correlation between the abiotic parameters themselves. DMS and DMSP had negative correlations with nutrients along the three transects during summer and winter except positive correlations between DMS and nutrients ($PO_4^{3-}$ and $SiO_3^{2-}$) along transect H19-26 during winter. In addition, positive correlations between DMS, DMSPd, and DMSPt along transect B57-63 and B12-17 during summer and positive correlation between DMSPt and Chl $a$ along transect B12-16 during winter indicated that DMSP was the phytoplankton-derived precursor of DMS (Table 2).

The AAd concentrations in the porewater of the surface sediments during summer were 13.52-136.42 μmol L$^{-1}$, with an average of $73.03 \pm 46.05$ μmol L$^{-1}$ (Table 3). However, no significant correlation was observed between the AAd concentrations in the porewater and those in the bottom seawater. The maximum concentration of AAd was observed at station H23; meanwhile, the AAd concentrations were all relatively high in the sediment porewater of transect H19-26 in the SYS, with an average of 121.79 μmol L$^{-1}$. The stations at transect H10-18 in the SYS and transect B12-17 in the NYS showed similar AAd concentrations (about 45 μmol L$^{-1}$), whereas the AA concentrations at stations (B61 and B63) in the BS showed big differences. Generally, the AAd concentrations in the porewater of the surface sediments were higher in the YS than in the BS.

### 3.3 Degradation of DMSPd and AAd in the BS and YS

The DMSPd and AAd degradation experiments were conducted using seawater at the endpoint stations of the investigated transects in the BS and YS during the two cruises. The production and/or degradation rates of DMSPd, DMS, and AAd are summarized in Table 4. In summer, the rates of DMS production were significantly lower than the rates of DMSPd degradation (Mann-Whitney test, $p = 0.01$) at all stations, whereas the rates of AAd production were slightly higher than the rates of DMSPd degradation at stations B12 and B63. The rates of AAd production were higher than those of DMS production (Mann-Whitney test, $p < 0.05$) at all stations. The enzymatic cleavage ratio of DMSP can be estimated using the ratio of the DMS production rate and the DMSPd degradation rate. The ratios were within the range of 7.8%-64.5%, with a mean of 27.7%. The maximum rates of DMSPd degradation ($5.76 \pm 0.47$ nmol L$^{-1}$ h$^{-1}$) and DMS ($2.71 \pm 0.36$ nmol L$^{-1}$ h$^{-1}$) and AAd ($5.20 \pm 0.40$ nmol L$^{-1}$ h$^{-1}$) production occurred at stations B57 and B63 in the BS, respectively. The minimum rates of DMS ($0.29 \pm 0.12$ nmol L$^{-1}$ h$^{-1}$) and AAd ($1.15 \pm 0.31$ nmol L$^{-1}$ h$^{-1}$) production occurred at stations H26 and H19 in the SYS, respectively. Although the rates of AAd microbial degradation at all stations were extremely high compared to the rates of AAd production and AAd photochemical degradation due to the addition of exogenous AAd at the beginning of incubation, the measured rates still reflect the capability of bacterially mediated degradation of AAd. Specifically, the AAd microbial degradation rates were higher at the inshore stations than the offshore stations, and the rates in the NYS were lower than those in the BS and the SYS. Moreover, the average AAd photochemical degradation rates were higher in the SYS than in the BS and the NYS. Since the DMSPd and AAd degradation follow first-order

kinetics (Kiene and Linn, 2000a; Wu et al., 2015), the turnover times of DMSPd and the rate constants of the AAd microbial and photochemical degradation were calculated (Table 4). The turnover times of DMSPd in the BS and YS fell in the range of 0.03-2.8 d which were estimated in earlier studies using radioisotopes, inhibitors, and low-level addition methods in different oceanic regions worldwide (Ledyard and Dacey, 1996; Kiene and Linn, 2000a; Simó et al., 2000). In addition, the AAd microbial degradation rate constants were higher than the AAd photochemical degradation rate constants at most stations.

Almost all degradation/production rates were lower in winter than in summer. Furthermore, the turnover times of DMSPd were much longer in winter than in summer (Mann-Whitney test, $p < 0.05$) but still fell in the range of earlier studies. The rates of DMS production were lower than the rates of DMSPd degradation and AAd production (Mann-Whitney test, $p < 0.05$) in winter, indicating an agreement with the results obtained in summer. Even though the difference in the DMS production rates between the stations was not large, the maximum rates of DMSPd degradation ($2.26 \pm 0.75$ nmol $L^{-1}$ $h^{-1}$), DMS production ($0.10 \pm 0.02$ nmol $L^{-1}$ $h^{-1}$), and AAd production ($1.48 \pm 0.29$ nmol $L^{-1}$ $h^{-1}$) were all observed in the SYS, which was different from the results obtained in summer. The enzymatic cleavage ratio of DMSP (3.5%-11.1%; average: 7.0%) was much lower in winter than in summer. The microbial degradation rates of AAd significantly decreased from summer to winter, but the rate constants in winter did not show a substantial decline compared to those in summer and even increased slightly at some stations. The AAd microbial degradation rates and rate constants were higher than the photochemical rates and rate constants at most stations in winter; this result was in agreement with that obtained in summer.

## 4 Discussion

### 4.1 Biogeochemical processes influencing the AAd in the surface water of the BS and YS

In summer, the average concentrations of $PO_4^{3-}$ in the BS (0.04 µmol $L^{-1}$), the NYS (0.05 µmol $L^{-1}$) and the SYS (0.04 µmol $L^{-1}$) were similar, however, the average $NO_3^-$, $NO_2^-$, and $SiO_3^{2-}$ concentrations in the BS ($NO_3^-$: 0.89 µmol $L^{-1}$; $NO_2^-$: 0.18 µmol $L^{-1}$; $SiO_3^{2-}$: 7.91 µmol $L^{-1}$) were much higher than those in the NYS ($NO_3^-$: 0.22 µmol $L^{-1}$; $NO_2^-$: 0.04 µmol $L^{-1}$; $SiO_3^{2-}$: 3.26 µmol $L^{-1}$) and the SYS ($NO_3^-$: 0.52 µmol $L^{-1}$; $NO_2^-$: 0.10 µmol $L^{-1}$; $SiO_3^{2-}$: 4.17 µmol $L^{-1}$). Therefore, the high total nutrient contents, which were attibuted to poor water circulation in the BS, promoted phytoplankton productivity and resulted in high Chl $a$ contents in the BS (Wei et al., 2004; Wang et al., 2009).The minimum value of Chl $a$ was observed in the center of the SYS and was ascribed to limited phytoplankton growth due to low nutrient contents (concentration of total inorganic nutrients < 3 µmol $L^{-1}$); the maximum value occurred in the southern area of the SYS and was due to high nutrient concentrations (total inorganic nutrients concentration of about 15 µmol $L^{-1}$) delivered via the CRDW (Wei et al., 2010).

The AAd concentrations in the BS and YS during summer were an order of magnitude higher than those (0.8-2.1 nmol $L^{-1}$, median 1.5 nmol $L^{-1}$) in the northern Gulf of Mexico in September 2011 (Tyssebotn et al., 2017). The reasons for these differences might be related to differences in sample storage, analytical methods and study areas. We stored the samples at 4 °C, whereas Tyssebotn et al. (2017) stored the samples at -20 °C. In addition, our study area was strongly affected by anthropogenic activities. Relatively higher AAd concentrations in the BS and the NYS than in the SYS during summer implied that terrestrial inputs might play an important role in controlling the AAd distribution in the BS and the NYS. It has been reported that the Yalu River flowing into the NYS has large amounts of organic pollutants, including AA (Liu, 2001); in addition, the densely populated Chengshan Cape may also be an anthropogenic source of AAd to the NYS. Furthermore, poor water circulation in the semi-enclosed NYS and inner BS favors local accumulations of AAd. On the contrary, the SYS is a relatively open water area and thus is much less affected by terrestrial discharges. Moreover,

AAd from DMSP degradation was not abundant in the SYS although the Chl $a$ values were relatively high, which might be related to the dominance of primary phytoplankton species with low ability of AAd production. Specifically, diatoms, a type of algal with low ability of DMSP and AAd production, were dominant in the SYS during summer (Liu et al., 2015). According to Zhang (2018), the maximum phytoplankton abundance in the SYS was 172.39 cell $mL^{-1}$, of which the diatom abundance accounted for 146.81 cell $mL^{-1}$. Furthermore, the diatom/dinoflagellate ratio was 28.96. In addition, some freshwater algae which do not produce DMSP and AAd, have been found adjacent to the Changjiang Estuary (Luan et al., 2006), and the north branch of the Changjiang Estuary flows into the SYS. All of these factors may have led to low AAd concentrations in the SYS.

The Chl $a$ contents were substantially lower in winter ($< 1$ µg $L^{-1}$ overall) than those in summer due to the lower temperature, light intensity, and phytoplankton activities, whereas the distribution patterns of Chl $a$ were similar in the two seasons. These results were in agreement with Zhang (2018), who found that the average phytoplankton abundance in winter (3.84 cell $mL^{-1}$) was much lower than that in summer (29.81 cell $mL^{-1}$), but diatoms (3.83 cell $mL^{-1}$) were still the dominant type of phytoplankton in winter. Moreover, Sun et al. (2001) also found that the diatoms in the study area consisted primarily of small diatoms in winter and larger diatoms in summer.

The AAd, DMS, and DMSP concentrations in the surface seawater during winter were about 2-4 times lower than those during summer (Table 1), but the distribution patterns were similar. Jin (2016) and Sun (2017) found significant positive correlations between DMS(P) and Chl $a$ during summer (DMS: $r = 0.418$, $n = 50$, $p < 0.01$; DMSPd: $r = 0.351$, $n = 50$, $p < 0.05$) and winter (DMS: $r = 0.629$, $p < 0.01$; particulate DMSP (DMSPp): $r = 0.527$, $p < 0.01$). These results demonstrated that DMS(P) originated primarily from biological production, which was stronger in summer than in winter. However, AAd showed no correlations with Chl $a$, nutrients, DMS, and DMSP in the entire study area during summer and winter; the reason may be that we only measured dissolved AA. It is assumed that the majority of AA produced from DMSPd degradation is stored intracellularly (Kinsey et al., 2016; Tyssebotn et al., 2017), whereas the majority of the produced DMS is found in the dissolved phase (Spiese et al., 2016). Therefore, AAd was not correlated with other biological parameters but DMS presented good correlations with others. In addition to biological production, terrestrial inputs might affect the AAd distributions. Therefore, AAd exhibited high values near the Chengshan Cape, which has intense human activities; in this area, Chl $a$, DMS, DMSP, and phytoplankton abundance also had high values. Nonetheless, the terrestrial inputs were weaker in winter than in summer, which resulted in slightly higher AAd concentrations in the BS than in the YS. AAd, DMS, and DMSP exhibited relatively high values in the BS and the NYS, and the concentrations decreased from the inshore to offshore areas in the SYS during summer and winter; these results were consistent with the distribution patterns in the BS and YS during autumn (Liu et al., 2016).

The positive correlation between AAd and temperature in the NYS during summer and in the BS during winter (Table 2) indicated that high temperatures might have enhanced both the biological production and the terrestrial sources of AAd. The positive correlation between AAd and DMSPd in the SYS during summer suggested that AAd in the SYS was mainly produced by DMSPd degradation rather than terrestrial inputs.

**4.2 Biogeochemical processes influencing AAd, DMS, and DMSP in the vertical profiles of the BS and YS**

In summer, the average concentration order was AAd > DMSPt > DMS > DMSPd along the three transects; this result was consistent with the order in the surface seawater (Table 1). Higher values of DMS than DMSPd might be produced through the intra-cellular cleavage of phytoplankton DMSPp catalyzed by the enzyme DMSP lyase and the photochemical and biological reduction of dimethylsulfoxide (DMSO) to DMS (Asher et al., 2017). In contrast, the higher values of AAd than DMSPt indicated that there were terrestrial sources of AAd aside from the contribution of in situ DMSP degradation along the three transects. Although there were only small differences in the average concentrations of sulfur compounds

between the three transects, the average concentrations of AAd showed significant differences (Kruskal-Wallis test, $p <$ 0.05). For instance, the AAd concentrations along transect B12-17 (NYS) and transect B57-63 (BS) were higher than those along transect H19-26 (SYS), which was in agreement with the distributions in the surface seawater. The high concentration could be ascribed to anthropogenic activities. The average contents of both Chl $a$ and DMSPt along the three transects followed the order: B57-63 > B12-17 > H19-26. This result suggested that large amounts of phytoplankton biomass might have induced high concentrations of DMSPt.

In winter, the average Chl $a$ and DMS concentrations along transect B12-16 were about twice as high as those along transect H19-26, which suggested that Chl $a$ had a controlling effect on DMS production. However, the average concentrations of DMSPd, DMSPt, and AAd along transect H19-26 were quite similar to those along transect B12-16; this result implied that the enzymatic cleavage of DMSP had been enhanced, and river discharges were not the dominant influence on the concentrations of AAd in winter. The concentration order along both transect H19-26 and transect B12-16 was AAd > DMSPt > DMSPd > DMS. The AAd concentrations were only slightly higher than the DMSPt concentrations, whereas the DMSPd concentrations exceeded the DMS concentrations in winter.

A comparison of the vertical profiles in different seasons (Figs. 4 and 5, Table 1) indicated that the DMS concentrations declined dramatically (by more than 5 nmol L$^{-1}$) from summer to winter, and the DMSPd concentrations also exhibited significant seasonal variations. The DMSPt concentrations were also slightly higher in summer than in winter, which was consistent with the seasonal pattern of Chl $a$, indicating the control of phytoplankton in DMS(P) production in both seasons. The higher AAd concentrations in summer than in winter were the combined result of high phytoplankton biomass and terrestrial inputs in summer. Overall, the reduced AAd concentrations from summer to winter along transect H19-26 were lower than those along transect B12-17(16), which suggested that terrestrial discharges contributed substantially to the AAd concentrations in the NYS and thus influenced the spatial distribution.

The AAd concentrations in the porewater were extremely higher in our study than those (50-60 nmol L$^{-1}$) in the Gulf of Mexico, as reported by Vairavamurthy et al. (1986). The differences might be attributed to differences in the sampling and analytical methods and the locations. In the study by Vairavamurthy et al. (1986), sediment porewater was obtained by centrifugation of thawed samples that were kept deep-frozen and the authors measured only two samples using electron capture gas chromatography, whereas we collected porewater via Rhizon soil moisture samplers connected to vacuum tubes and analyzed samples using high-performance liquid chromatography. The pressure in the vacuum tube might have caused cell breakage in the sediments, thus releasing large amounts of AAd in the porewater. Moreover, the bacteria abundance and species in the sediments of the BS and YS in 2015 might be different from those in the Gulf of Mexico in 1986. Wang (2015) reported that δ- and γ-proteobacteria were the dominant taxa in the sediments of the BS and YS, with proportions in the range of 24%-70%. DddY, which is the only known periplasmic DMSP lyase (Li et al., 2017), is widely present in δ- and γ-proteobacteria and can cleave large amounts of intracellular DMSP (mmol L$^{-1}$ levels) concentrated by DMSP catabolizing bacteria (Wang et al., 2017). Therefore, all those factors led to high AAd concentrations in the porewater of the surface sediments.

Slezak et al. (1994) discovered that the bacterial activity was reduced at AA concentrations > 10 µmol L$^{-1}$ in long-term incubations of seawater cultures (24 to 110 h). Therefore, AAd in the porewater might have reduced the bacterial metabolism, thus impacting the microbial community in the sediments; this aspect is very important in the study of marine sediment ecosystems. In addition, we speculated that high concentrations of AAd in the sediments might have been transported to the bottom seawater because Nedwell et al. (1994) found that DMS was emitted to the water column from the sediments. To date, there are very few studies on AAd in sediments, and the potential factors influencing AAd concentrations in porewater remain unknown. For a better understanding of the source and fate of AAd in marine

sediments, a detailed investigation of multiple parameters such as dissolved organic carbon, DMS, and DMSP in sediments is needed.

**4.3 Degradation of DMSPd and AAd in the BS and YS**

The microbial degradation rates of AAd in the BS and YS during summer were extremely higher than the total biological uptake of AAd (0.07-1.8 nmol $L^{-1}$ $d^{-1}$) in the northern Gulf of Mexico in September 2011 (Tyssebotn et al., 2017); these discrepancies might be due to differences in the initial concentrations. Specifically, in our study, we added exogenous AAd at the beginning of incubation. Nevertheless, we found that the microbial degradation rates were higher at the inshore stations than the offshore stations. In addition, almost all production/degradation rates during summer and winter were independent of Chl *a*; these results were consistent with the results of Motard-Côté et al. (2016) and Tyssebotn et al. (2017) .

The production/degradation rates of DMSPd, DMS, and AAd exhibited similar distributions in different sea areas during different seasons. For instance, the DMS production rates were lower than the AAd production rates at all stations in both summer and winter, implying that AAd was produced by DMSP through more complicated demethylation processes in addition to enzymatic cleavage, which is thought to be the sole pathway of DMS production from DMSP. The low enzymatic cleavage ratio (<50%) during both summer and winter indicated that the enzymatic cleavage was not the dominant pathway of DMSP degradation (Ledyard and Dacey, 1996; Kiene and Linn, 2000b). It is noteworthy that the AAd production rates were slightly higher than the DMSPd degradation rates at some stations during summer and winter; the reason might be the direct production from DMSPp at those stations in addition to the exogenous DMSPd during the incubation experiments. In addition, the AAd microbial degradation rates were always higher than the photochemical degradation rates, suggesting that microbial degradation was a more important pathway of AAd removal than photochemical degradation.

Nevertheless, the production/degradation rates of DMSPd, DMS, and AAd also showed seasonal and spatial variations. Higher production/degradation rates of DMSPd, DMS, and AAd in summer than in winter indicated that the temperature promoted the degradation/production rates. In addition, the seasonal differences in bacteria abundance and light intensity also made great contributions to different rates of microbial degradation and photochemical degradation, respectively. According to Liang et al. (2019), the abundances of *Vibrio* ($\gamma$-proteobacteria) averaged $1.4\times10^6$ copies $L^{-1}$ in summer, which was significantly higher than in winter (mean value of $1.9\times10^5$ copies $L^{-1}$) (Mann-Whitney test, $p < 0.01$). Significant seasonal differences in total bacterial abundance were also observed (Mann-Whitney test, $p < 0.001$). The average light intensity in summer was 49400 lx, which was higher than that in winter (34050 lx). All those factors led to high degradation/production rates in summer. In addition, Liang et al. (2019) also found that the dominant bacteria groups exhibited different distributions in abundance with different seasons and sea areas. Specifically, the abundance of *V. campbellii* was higher in the YS than in the BS in summer ($p < 0.05$), whereas the abundance of *V. caribbeanicus* drastically decreased from the BS to the YS ($p < 0.05$). Therefore, the different microbial degradation/production rates of DMSPd, DMS, and AAd in different sea areas might have resulted from the differences in bacteria species and abundance in the BS and YS. Moreover, there are differences in the capabilities of different bacteria species to degrade AAd, which resulted in the disparities of AAd microbial degradation rates and rate constants between the inshore and offshore stations. We applied a simple box model to estimate the contribution of different sources and sinks of AAd in the surface seawater of the BS and YS:

$$dc/dt = r_{prod} - r_{bio} - r_{photo} + r_{other}$$

We assumed that AAd concentrations were in a steady state; therefore, $dc/dt = 0$. The AAd production rate ($r_{prod}$) was calculated by multiplying the AAd production rate constant with in situ concentration. The AAd microbial degradation

rate ($r_{bio}$) and photochemical degradation rate ($r_{photo}$) were calculated similarly. $r_{other}$ represented sources and sinks of AAd other than the production from DMSPd. Based on the equations, the mean $r_{prod}$, $r_{bio}$, and $r_{photo}$ in summer were 5.76 nmol L$^{-1}$ h$^{-1}$, 8.43 nmol L$^{-1}$ h$^{-1}$, and 2.83 nmol L$^{-1}$ h$^{-1}$, respectively; the results indicated that there were other sources of AAd, i.e., a production rate of 5.50 nmol L$^{-1}$ h$^{-1}$. These sources might include the production from DMSPp, riverine inputs and other unknown sources. In winter, the mean $r_{prod}$, $r_{bio}$, and $r_{photo}$ were 1.65 nmol L$^{-1}$ h$^{-1}$, 2.66 nmol L$^{-1}$ h$^{-1}$, and 1.32 nmol L$^{-1}$ h$^{-1}$, respectively, and the rate from other sources was 2.33 nmol L$^{-1}$ h$^{-1}$. The relationship of the rates from other sources between summer and winter was similar to that of the AAd concentrations in the surface seawater between summer and winter; namely, the rate from the other sources and the AAd concentrations in the surface seawater in winter were less than half of those in summer.

## 5 Conclusions

We investigated the horizontal and vertical distributions of AAd, DMS, and DMSP in the BS and YS during summer and winter. Significant seasonal variations were observed in the study area. The AAd concentrations were relatively higher in the surface seawater during summer than during winter due to strong biological production from DMSP and abundant terrestrial inputs from rivers in summer. The distribution patterns of AAd were similar during summer and winter, i.e., relatively high values of AAd occurred in the BS and the NYS, and the concentrations decreased from the inshore to offshore areas in the SYS. In the vertical profiles, high values of AAd, DMS, and DMSP were mostly observed in the upper layers with occasional high values in the bottom layers along the three different transects. The average concentration sequence was AAd > DMSPt > DMS > DMSPd among all three transects during summer, illustrating that DMSPp acted as a DMS producer and terrestrial sources of AAd were present. In contrast, the sequence along transects in winter was AAd > DMSPt > DMSPd > DMS. DMS and AAd presented a stronger decrease from summer to winter than DMSP along transects. We also measured the AAd concentrations in the porewater of the surface sediments. The extremely high AAd concentrations in the porewater were attributed to the abundant bacteria and active bacteria DMSP lyases in the sediments. Moreover, the DMS and AAd production from DMSPd degradation and the AAd degradation rates were always higher during summer than during winter. The AAd microbial degradation rates and rate constants were higher than the photochemical degradation rates and rate constants during both summer and winter. The AAd production and degradation experiments also proved that other sources of AAd existed in addition to the production from DMSPd.

**Data availability.** Most data are shown in figures, tables and references. More detailed data can be accessed by email request to the corresponding authors.

**Author contribution.** Xi Wu participated those two cruises to collect and analyse samples. Xi Wu and Chun-Ying Liu designed the on-deck experiments and Xi Wu carried them out. Xi Wu prepared the manuscript with contributions from all co-authors.

**Competing interests.** The authors declare that they have no conflict of interest.

**Acknowledgments.** We thank the captain and crew of the R/V "Dong Fang Hong 2" for their help during the investigations. We also thank the associate editor and two anonymous reviewers for their constructive comments, which greatly improved the manuscript.

**Financial support.** This work was financially supported by the National Key Research and Development Program of China (No. 2016YFA0601301), the National Natural Science Foundation of China (Nos. 41676065 and 41176062) and the Fundamental Research Funds for the Central Universities (No. 201762032).

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

 **Figure Captions**

**Figure 1: Locations of the sampling stations in the BS and YS during summer (a) and winter (b). (c) Schematic circulations and water masses in the BS and YS (Su, 1998; Lee et al., 2000). BSCC: Bohai Sea Coastal Current; YSCC: Yellow Sea Coastal Current; KCC: Korea Coastal Current; YSWC: Yellow Sea Warm Current; CRDW: Changjiang River Diluted Water; YSCWM: Yellow Sea Cold Water Mass.**

 **Fig. 2. Horizontal distributions of Chl *a* (µg L$^{-1}$) and AAd (nmol L$^{-1}$) in the surface water of the BS and YS during summer and winter. a: Chl *a* in summer; b: AAd in summer; c: Chl *a* in winter; d: AAd in winter.**

**Fig. 3. Horizontal distributions of DMS (nmol L$^{-1}$), DMSPd (nmol L$^{-1}$), and DMSPp (nmol L$^{-1}$) in the surface water of the BS and YS during summer and winter. Data in summer and winter presented here were described by Jin (2016) and Sun (2017) respectively.**

 **Fig. 4. Vertical profiles of temperature (°C), Chl *a* (µg L$^{-1}$), AAd (nmol L$^{-1}$), DMS (nmol L$^{-1}$), DMSPd (nmol L$^{-1}$), and DMSPt (nmol L$^{-1}$) along transect B57-63, transect B12-17, and transect H19-26 during summer. Kriging method is used for interpolating contours. The black dots represent sampling points.**

**Fig. 5. Vertical profiles of temperature (°C), Chl *a* (µg L$^{-1}$), AAd (nmol L$^{-1}$), DMS (nmol L$^{-1}$), DMSPd (nmol L$^{-1}$), and DMSPt (nmol L$^{-1}$) along transect B12-16 and transect H19-26 during winter. Kriging method is used for interpolating contours. The**
 **black dots represent sampling points.**

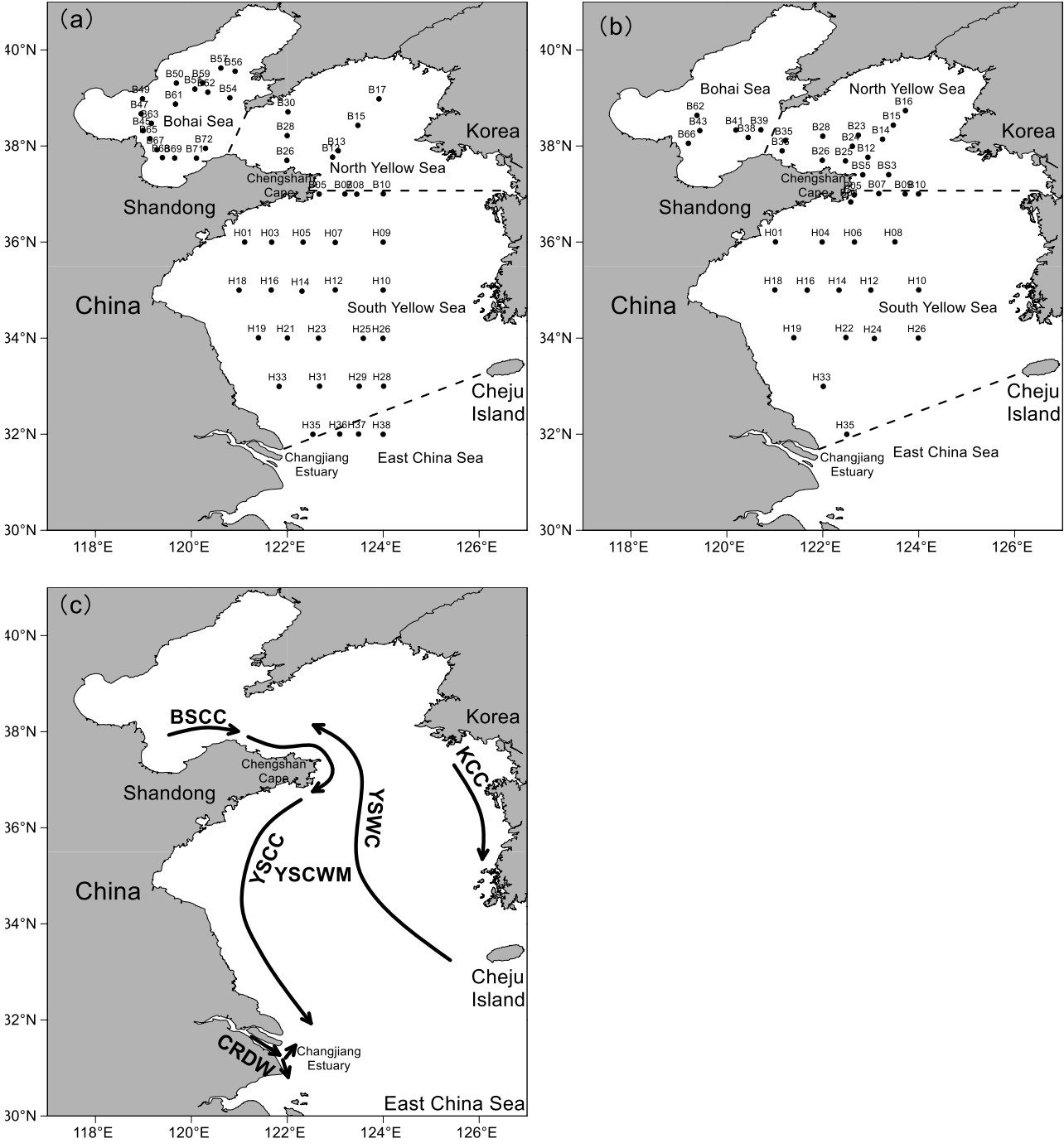

Fig. 1. Locations of the sampling stations in the BS and YS during summer (a) and winter (b). (c) Schematic circulations and water masses in the BS and YS (Su, 1998; Lee et al., 2000). BSCC: Bohai Sea Coastal Current; YSCC: Yellow Sea Coastal Current; KCC: Korea Coastal Current; YSWC: Yellow Sea Warm Current; CRDW: Changjiang River Diluted Water; YSCWM: Yellow Sea Cold Water Mass.

**Summer**

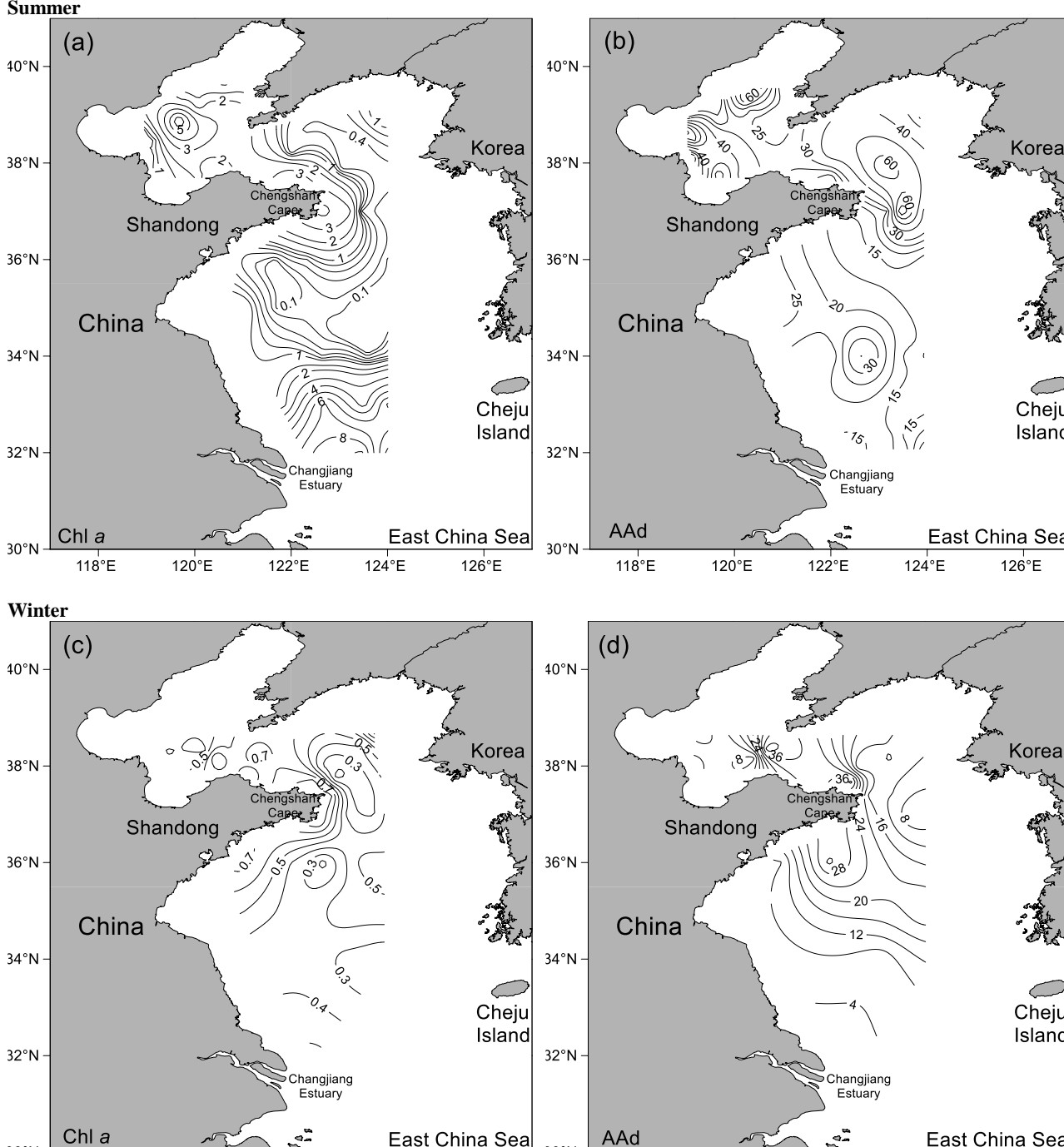

**Fig. 2. Horizontal distributions of Chl *a* (µg L⁻¹) and AAd (nmol L⁻¹) in the surface water of the BS and YS during summer and winter. (a): Chl *a* in summer; (b): AAd in summer; (c): Chl *a* in winter; (d): AAd in winter.**

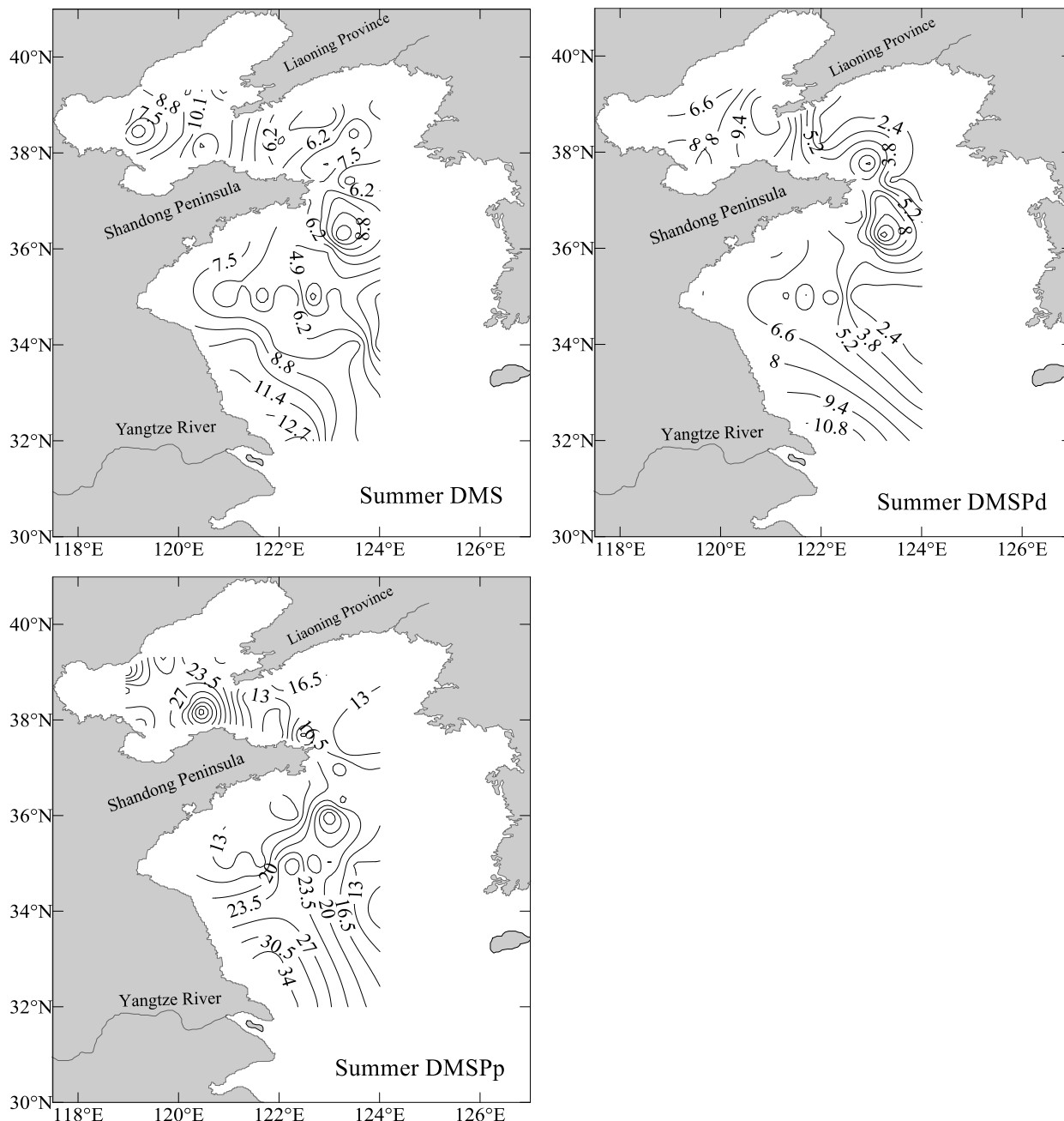

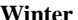

Fig. 3. Horizontal distributions of DMS (nmol L$^{-1}$), DMSPd (nmol L$^{-1}$), and DMSPp (nmol L$^{-1}$) in the surface water of the BS and YS during summer and winter. Data in summer and winter presented here were described by Jin (2016) and Sun (2017) respectively.


**Transect B57-63**

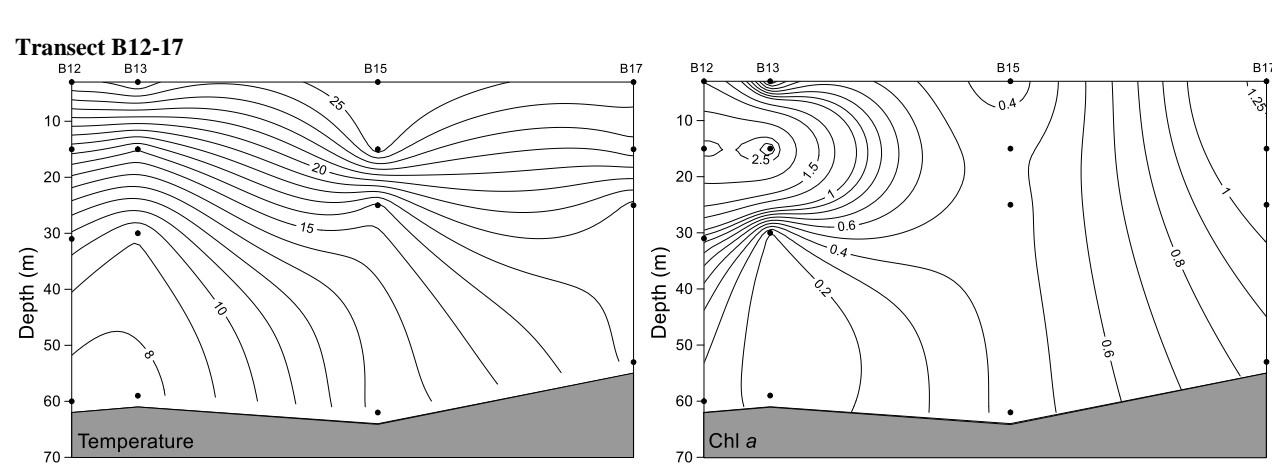

**Transect B12-17**

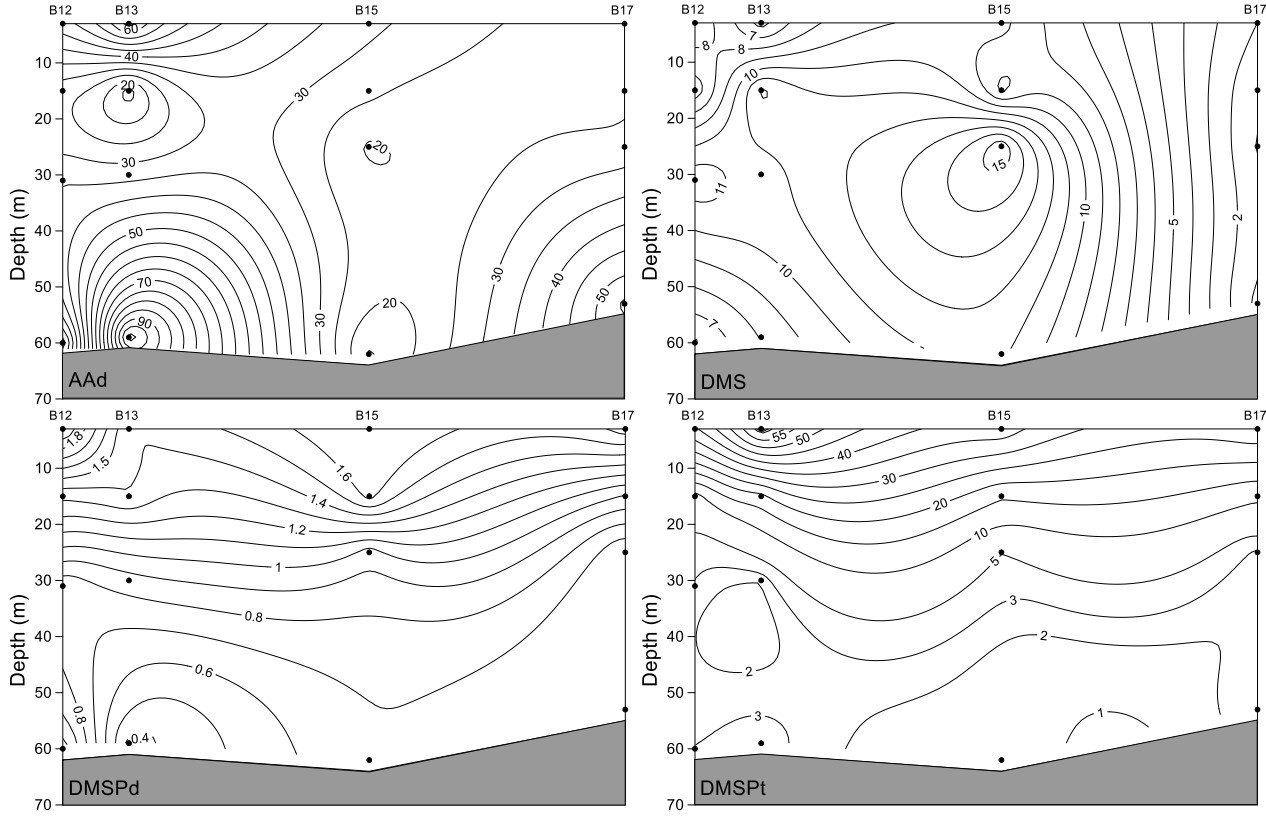

**Transect H19-26**

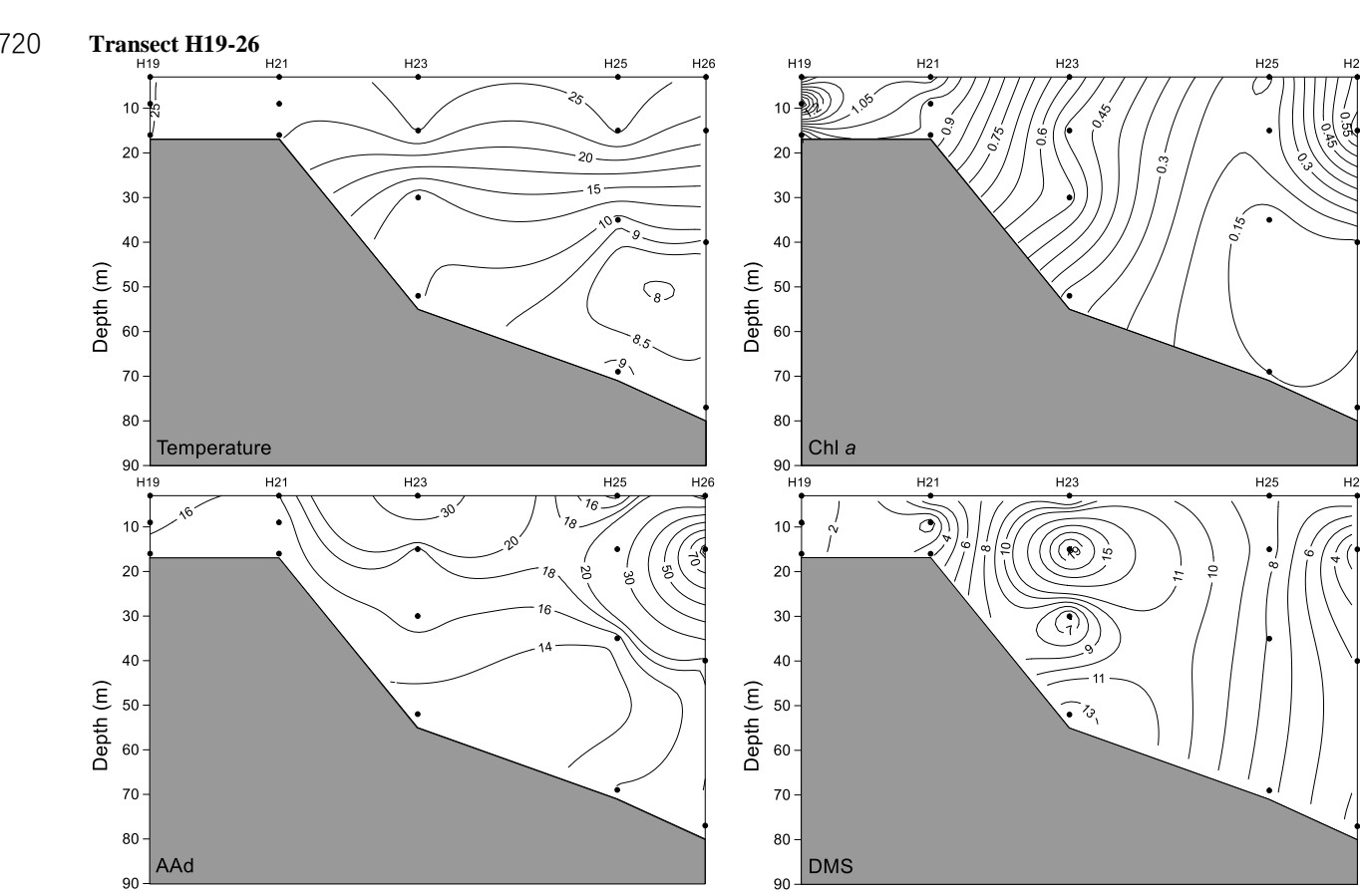

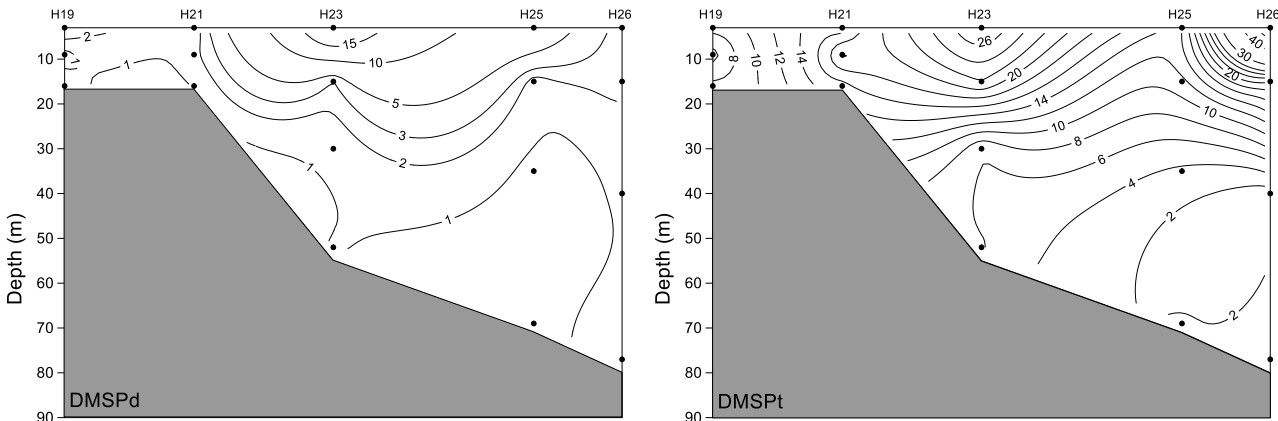

**Fig. 4. Vertical profiles of temperature (°C), Chl *a* (µg L⁻¹), AAd (nmol L⁻¹), DMS (nmol L⁻¹), DMSPd (nmol L⁻¹), and DMSPt (nmol L⁻¹) along transect B57-63, transect B12-17, and transect H19-26 during summer. Kriging method is used for interpolating contours. The black dots represent sampling points.**


**Transect B12-16**




**Transect H19-26**

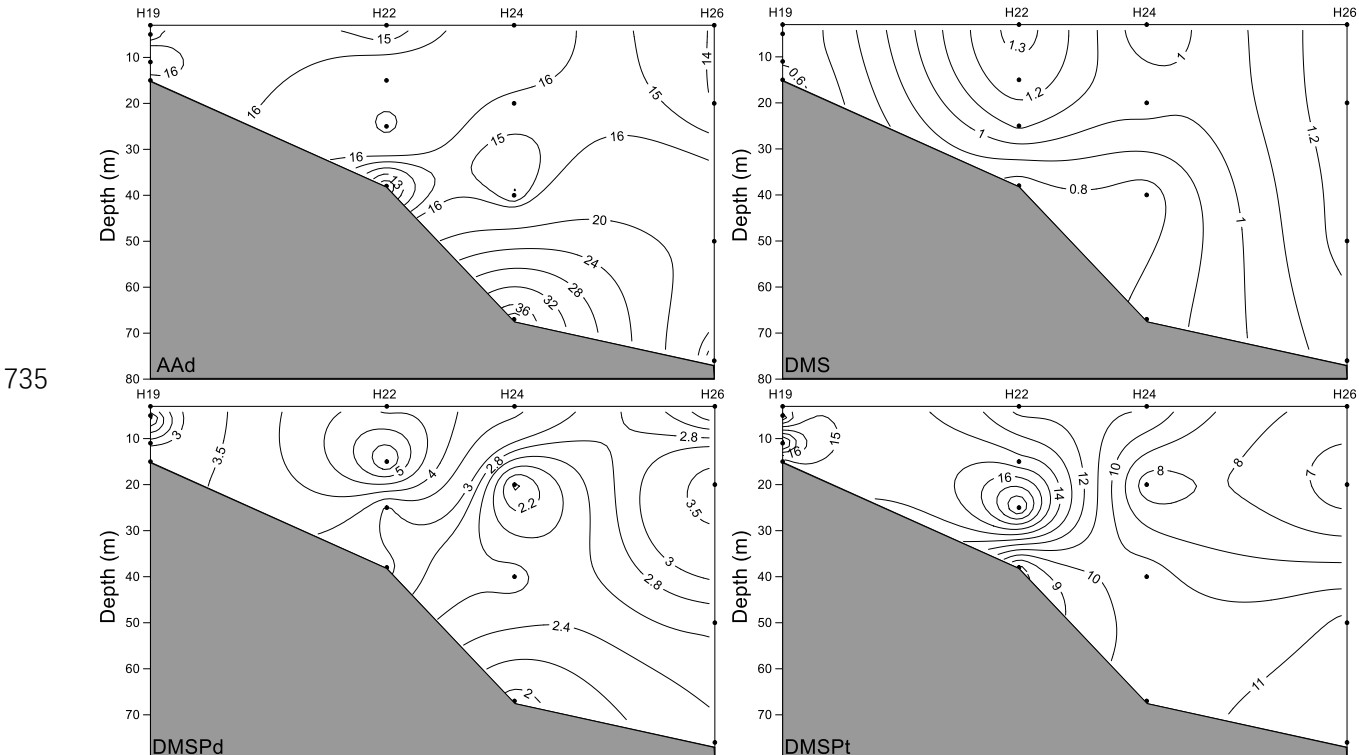

**Fig. 5. Vertical profiles of temperature (°C), Chl *a* (μg L$^{-1}$), AAd (nmol L$^{-1}$), DMS (nmol L$^{-1}$), DMSPd (nmol L$^{-1}$), and DMSPt (nmol L$^{-1}$) along transect B12-16 and transect H19-26 during winter. Kriging method is used for interpolating contours. The black dots represent sampling points.**

**Table Captions**

**Table 1** Summary of the mean values (ranges) and the significance of seasonal differences of AAd, DMS, DMSPd, and DMSPt in the surface seawater of the BS and YS and the entire vertical profiles of transects during summer and winter. The significance of seasonal differences was obtained using Mann-Whitney test.

**Table 2** Correlations between AAd, DMS, DMSP, and other biogeochemical parameters in the BS and YS during summer and winter. Pearson correlation test was used here.

**Table 3** The AAd concentrations in the porewater of the surface sediments and in the bottom seawater during summer 2015.

**Table 4** Rates and rate constants of DMS and AAd production from DMSPd degradation and AAd degradation in the BS and YS during summer and winter.

**Table 1** Summary of the mean values (ranges) and the significance of seasonal differences of AAd, DMS, DMSPd, and DMSPt in the surface seawater of the BS and YS and the entire vertical profiles of transects during summer and winter. The significance of seasonal differences was obtained using Mann-Whitney test.

| | | AAd (nmol L$^{-1}$) | DMS (nmol L$^{-1}$) | DMSPd (nmol L$^{-1}$) | DMSPt (nmol L$^{-1}$) |
|---|---|---|---|---|---|
| Summer | Surface | 30.01 ± 21.12 (10.53-92.29) | 6.12 ± 3.01 (1.10-14.32)* | 6.03 ± 3.45 (1.05-13.23)* | 28.86 ± 14.15 (8.70-63.03)* |
| | B57-63 | 36.36 ± 23.57 (11.08-73.06) | 5.51 ± 2.01 (2.57-8.79) | 1.56 ± 0.84 (0.72-3.37) | 22.94 ± 21.28 (4.12-56.61) |
| | B12-17 | 34.60 ± 26.00 (12.77-102.98) | 7.37 ± 4.50 (0.74-15.76) | 1.12 ± 0.48 (0.36-2.01) | 15.45 ± 17.98 (1.90-63.03) |
| | H19-26 | 22.24 ± 18.25 (13.19-85.86) | 6.44 ± 5.14 (0.79-21.98) | 3.05 ± 4.92 (0.61-21.59) | 13.67 ± 12.90 (1.11-55.14) |
| Winter | Surface | 14.98 ± 7.22 (4.28-42.05) | 1.38 ± 0.41 (0.54-2.22)* | 2.30 ± 0.80 (1.16-4.29)* | 10.39 ± 4.14 (2.36-22.21)* |
| | B12-16 | 17.68 ± 5.21 (13.94-27.69) | 1.99 ± 1.02 (1.12-4.56) | 2.92 ± 0.82 (1.54-4.55) | 11.44 ± 5.89 (5.33-24.50) |
| | H19-26 | 17.08 ± 6.72 (11.04-39.47) | 0.96 ± 0.29 (0.52-1.35) | 3.06 ± 1.07 (1.92-6.06) | 11.88 ± 3.97 (6.12-19.92) |
| Seasonal difference | Surface | $p < 0.001$ | $p < 0.001$ | $p < 0.01$ | $p < 0.001$ |
| | B12-16 | $p < 0.05$ | $p < 0.05$ | $p < 0.001$ | |
| | H19-26 | | $p < 0.001$ | $p < 0.01$ | |

* collected from published MS theses (Jin, 2016; Sun, 2017)

**Table 2 Correlations between AAd, DMS, DMSP, and other biogeochemical parameters in the BS and YS during summer and winter. Pearson correlation test was used here.**

| | | | T | S | Chl $a$ | DMS | DMSPd | DMSPt | AAd | $PO_4^{3-}$ | $SiO_3^{2-}$ | $NO_3^-$ | $NO_2^-$ | $NH_4^+$ |
|---|---|---|---|---|---|---|---|---|---|---|---|---|---|---|
| Summer | NYS surface | AAd | 0.676* | | | | | | | | | | | |
| | SYS surface | AAd | | | | | 0.626* | | | | | | | |
| | H19-26 | DMSPt | 0.549* | -0.555* | | | | | | | -0.486* | -0.510* | -0.510* | |
| | B12-17 | DMSPd | 0.742*** | -0.626** | | | | | | -0.745** | -0.737** | -0.784*** | -0.792*** | |
| | | DMSPt | 0.746*** | -0.707** | | | 0.725** | | | -0.630** | -0.850*** | -0.721** | -0.730** | |
| | | DMS | | | | | | | | | -0.619* | | | |
| | B57-63 | DMSPd | 0.593* | -0.843*** | | | | | | -0.806** | | | | |
| | | DMSPt | | -0.867*** | | 0.577* | 0.745** | | | -0.762** | | -0.650* | -0.647* | |
| Winter | BS surface | AAd | 0.972* | | | | | | | | | | | |
| | H19-26 | DMS | 0.765*** | 0.691** | | | | | | 0.772** | 0.824** | | | |
| | | DMSPt | -0.605* | -0.618* | | | | | | | | | | |
| | B12-16 | DMS | -0.859*** | -0.807** | | | | | | -0.670* | | | | |
| | | DMSPd | | | | | | | | -0.748* | | | | |
| | | DMSPt | | | 0.930*** | | | | | | -0.852** | | | |

*Significant at $p < 0.05$.
**Significant at $p < 0.01$.
***Significant at $p < 0.001$.

**Table 3 The AAd concentrations in the porewater of the surface sediments and in the bottom seawater during summer 2015.**

| Station | H10 | H12 | H14 | H16 | H19 | H23 | H25 | H26 | B12 | B13 | B61 | B63 |
|---|---|---|---|---|---|---|---|---|---|---|---|---|
| Sampling time | 08-19 06:59 | 08-19 15:28 | 08-19 21:48 | 08-20 03:11 | 08-20 14:35 | 08-21 00:21 | 08-21 08:03 | 08-21 11:24 | 08-28 17:20 | 08-28 19:58 | 09-02 14:42 | 09-02 19:54 |
| Porewater AAd ($\mu$mol L$^{-1}$) | 34.54 | 13.52 | 99.89 | 38.36 | 128.61 | 136.42 | 99.45 | 122.68 | 41.31 | 46.50 | 15.63 | 102.40 |
| Bottom AAd (nmol L$^{-1}$) | 14.34 | 13.41 | 12.32 | 17.54 | 15.59 | 13.25 | 16.23 | 19.01 | 16.74 | 102.98 | 18.95 | 23.68 |

**Table 4 Rates and rate constants of DMS and AAd production from DMSPd degradation and AAd degradation in the BS and YS during summer and winter.**

Summer

| Stations | SYS | | NYS | | BS | |
|---|---|---|---|---|---|---|
| | H19 | H26 | B12 | B17 | B57 | B63 |
| DMSPd degradation rates (nmol L$^{-1}$ h$^{-1}$) | 3.12 ± 0.69 | 3.72 ± 0.28 | 1.44 ± 0.39 | 1.83 ± 1.08 | 5.76 ± 0.47 | 4.20 ± 0.36 |
| DMSPd turnover times (h) | 6.25 | 5.10 | 19.31 | 14.29 | 4.91 | 5.88 |
| DMS production rates (nmol L$^{-1}$ h$^{-1}$) | 0.55 ± 0.32 | 0.29 ± 0.12 | 0.33 ± 0.05 | 0.69 ± 0.09 | 0.90 ± 0.46 | 2.71 ± 0.36 |
| AAd production rates (nmol L$^{-1}$ h$^{-1}$) | 1.15 ± 0.31 | 1.90 ± 0.61 | 2.53 ± 0.64 | 1.15 ± 0.69 | 2.63 ± 0.35 | 5.20 ± 0.40 |
| AAd microbial degradation rates (nmol L$^{-1}$ h$^{-1}$) | 25.36 ± 13.15 | 22.10 ± 0.89 | 15.07 ± 0.52 | 11.84 ± 0.45 | 16.17 ± 0.52 | 24.92 ± 3.18 |
| AAd photochemical degradation rates (nmol L$^{-1}$ h$^{-1}$) | 3.16 ± 0.36 | 3.45 ± 2.08 | 0.91 ± 0.16 | 4.02 ± 0.34 | 0.67 ± 0.09 | 2.36 ± 0.14 |
| AAd microbial degradation rate constants (h$^{-1}$) | 0.07 ± 0.05 | 0.36 ± 0.25 | 0.07 ± 0.004 | 0.30 ± 0.02 | 0.50 ± 0.03 | 0.03 ± 0.005 |
| AAd photochemical degradation rate constants (h$^{-1}$) | 0.01 ± 0.009 | 0.02 ± 0.03 | 0.03 ± 0.006 | 0.14 ± 0.01 | 0.04 ± 0.005 | 0.12 ± 0.007 |

Winter

| Stations | SYS | | NYS | |
|---|---|---|---|---|
| | H19 | H26 | B12 | B16 |
| DMSPd degradation rates (nmol L$^{-1}$ h$^{-1}$) | 2.26 ± 0.75 | 1.14 ± 0.50 | 1.92 ± 0.87 | 0.63 ± 0.59 |
| DMSPd turnover times (h) | 16.53 | 39.68 | 31.55 | 46.73 |
| DMS production rates (nmol L$^{-1}$ h$^{-1}$) | 0.08 ± 0.03 | 0.10 ± 0.02 | 0.09 ± 0.01 | 0.07 ± 0.05 |
| AAd production rates (nmol L$^{-1}$ h$^{-1}$) | 1.48 ± 0.29 | 1.22 ± 0.28 | 0.30 ± 0.25 | 0.91 ± 0.02 |
| AAd microbial degradation rates (nmol L$^{-1}$ h$^{-1}$) | 9.41 ± 0.59 | 4.73 ± 0.53 | 8.54 ± 0.08 | 18.66 ± 0.81 |
| AAd photochemical degradation rates (nmol L$^{-1}$ h$^{-1}$) | 4.30 ± 0.14 | 2.31 ± 0.48 | 2.72 ± 0.21 | 0.97 ± 0.46 |
| AAd microbial degradation rate constants (h$^{-1}$) | 0.06 ± 0.01 | 0.36 ± 0.07 | 0.18 ± 0.002 | 0.29 ± 0.02 |
| AAd photochemical degradation rate constants (h$^{-1}$) | 0.13 ± 0.005 | 0.06 ± 0.02 | 0.13 ± 0.01 | 0.05 ± 0.02 |
