# Peer review of "Acrylic acid and related dimethylated sulfur compounds in the Bohai and Yellow Seas during summer and winter"

_Biogeosciences, 2019_

## Referee Comment (RC1) · Anonymous Referee #1 · 4 Jul 2019

Review of BG-2019-172 by Wu et al. This paper describes the DMS/P and AA surface ocean cycling in the Bohai and Yellow Seas during winter and summer. The authors also measured depth profiles and porewater concentrations, as well as performed incubation experiments to derive production/degradation rates. This paper contains valuable data, but only a small amount of new science. By now, the community has a generally good understanding of DMS dynamics and the controlling factors. We know that phytoplankton, bacteria, and environmental parameters influence DMS/P cycling (and related compounds). Nonetheless, it appears that the authors did not measure phytoplankton or bacterial parameters. They attempt to explain processes without measuring the parameters involved. This paper is generally more suited to a journal like ESSD.

[Figure]

Specific comments: The English throughout the entire manuscript needs to be slightly revised. Overall, the language is fine, but there are still many mistakes. Section 2.3 Were there particulate measurements of anything (no filtering to measure total DMS/P, etc.)? Did you measure duplicates or triplicates? How exactly was precision and the limit of detection determined? Section 2/3 Were nutrients measured? Phytoplankton pigments or flow cytometry? Section 3.5 Were bacterial parameters measured? Why not? Did you see evidence of first order rates? Did you discover something new with the incubation experiments?

---

## Author Comment (AC1) · 11 Jul 2019

Review of BG-2019-172 by Wu et al. This paper describes the DMS/P and AA surface ocean cycling in the Bohai and Yellow Seas during winter and summer. The authors also measured depth profiles and porewater concentrations, as well as performed incubation experiments to derive production/degradation rates. This paper contains valuable data, but only a small amount of new science. By now, the community has a generally good understanding of DMS dynamics and the controlling factors. We know that phytoplankton, bacteria, and environmental parameters influence DMS/P cycling (and related compounds). Nonetheless, it appears that the authors did not measure

phytoplankton or bacterial parameters. They attempt to explain processes without measuring the parameters involved. This paper is generally more suited to a journal like ESSD.

We found phytoplankton and bacterial data of these two cruises in two published papers (Liang et al., 2019; Zhang 2018). We will explore how these parameters influence DMS/P cycling in revised manuscript. In addition, our study proved other sources of AA (e.g. terrestrial inputs from rivers and production from DMSPp) in surface seawater through on-deck incubation experiments. Although some observations and studies on the distributions of DMS and DMSP in the Bohai Sea and the Yellow Sea have been conducted (Li et al., 2016; Yang et al., 2014; 2015), the study aiming at winter has not been reported as well as the relationship between AA and DMS/P, which could reflect if temperature was a key controlling factor on biogenic sulfur cycling. Furthermore, our study was the first time to collect AA samples in porewater in Chinese marginal seas, although more work needs to be done to further understand the source and fate of AA in marine sediments. We will strength these particularities of our study in Section 1.

Specific comments: 1. The English throughout the entire manuscript needs to be slightly revised. Overall, the language is fine, but there are still many mistakes.

Thanks for your suggestions. We will check the entire manuscript to polish it and correct mistakes.

2. Section 2.3 Were there particulate measurements of anything (no filtering to measure total DMS/P, etc.)? Did you measure duplicates or triplicates? How exactly was precision and the limit of detection determined?

We measured total DMSP (DMSPt, no filtering) and dissolved DMSP (DMSPd, filtering with 0.7 $\mu$m GF/F). We did not measure particulate DMSP (DMSPp) directly, but DMSPp can be calculated using DMSPt minus DMSPd. Duplicates were measured. According to Kiene and Service (1991), precision was determined as following: DMS standards prepared in glycol were compared to DMSP standards which were

treated with base to produce DMS. This comparison between the two different standards showed agreement to within 5%. For typical water volumes analyzed (2 ml), this method gave detection limits for DMS of 0.05-0.5 nmol L-1. The limit of detection (LOD) may be expressed as: LOD =3*s/m. s is the standard deviation of low concentration samples and m is the slope of the calibration curve.

3. Section 2.3 Were nutrients measured? Phytoplankton pigments or flow cytometry?

Nutrients were measured. We will add analytical procedures in Section 2.3 and discuss how nutrients affect the distributions of DMS/P and AA in section 3. Utermöhl method was used for phytoplankton counting in Zhang's thesis (2018).

4. Section 3.5 Were bacterial parameters measured? Why not? Did you see evidence of first order rates? Did you discover something new with the incubation experiments?

We are sorry for not measuring the bacterial parameters, but we find a published paper (Liang et al., 2019) discussing the bacterial parameters of the same cruises. We will use this data to support our experiments and cite this paper in revised manuscript. In our published paper (Wu et al., 2015) which studied the acrylic acid (AA) degradation in details, we did incubation experiments for 8h and sampled every 2h. We found that AA degraded quickly in first 2h and the degradation rates reduced gradually, the loss curves fit the first-order equation. Kiene (1996) also demonstrated that apparent first order rate constants (k) for the loss of DMSPd were estimated by plotting the natural log of the DMSPd concentration vs time. Besides the DMSPd degradation experiments, we carried out the AA biological and photochemical degradation experiments simultaneously. We found the total consumption (biological + photochemical) rates of AA were always higher than the production rates of AA from DMSPd at different stations during these two cruises, which provided evidence for other sources of AA in this study area.

The following references are added.

Liang, J., Liu, J., Wang, X., Lin, H., Liu, J., Zhou, S., Sun, H., and Zhang, X.-H.:

Spatiotemporal dynamics of free-living and particle-associated Vibrio communities in the northern Chinese marginal seas, Appl. Environ. Microbiol., 85, e00217-00219, 2019. Zhang, D.: The study of phytoplankton and biosilicon in the Yellow Sea and the Bohai Sea (in Chinese with English abstract), MS thesis, ‎Tianjin University of Science & Technology, Tianjin, China, 2018.

---

## Referee Comment (RC2) · Anonymous Referee #2 · 14 Aug 2019

Wu et al. measured DMS(P) and AA concentrations across different oceanographic regimes, depths and seasons, rate measurements of DMS(P) and AA degradation and production, and AA concentrations in porewaters. AA is a product of DMSP cleavage and potentially an important carbon source, but little is known about global AA dynamics. I commend the authors for their expansive assessment of AA dynamics in the context of DMS(P) cycling. *These measurements reflect an important contribution to knowledge about AA's role in the marine microbial ecosystem.* However, the current manuscript requires significant improvements for accuracy and presentation clarity. Specifically:

1. Statistical tests are missing/incomplete throughout the manuscript. Any conclusions deemed significant should be supported by statistics. Overall, results should be made more quantitative.

2. More biological measurements are necessary to support conclusions. Only Chla concentrations are reported, which is well established to be a poor predictor of DMS(P) concentrations. This is not a focus here as authors have already reported they can add more biological parameters.

3. There are a significant number of citation errors in (both missing and incorrect citations) and I highly suggest the authors review their citations fully before resubmitting. Additionally, many conclusions are "overstated", meaning the strength of the wording should be edited.

4. The clarity of the manuscript would greatly benefit from dividing the Results into Results and Discussion. As it reads now, the results for each section are being explained in pieces but no full story of all the results is tied together.

5. Finally, the motivation of the manuscript should be clearer. I fully recognize that these measurements of AA will improve knowledge, but why is it important to fill that gap? What unknowns do these results answer about AA cycling? Given the expansive AA measurements, this manuscript could test more specific hypotheses. Additionally, for writing clarity, I would recommend focusing the questions towards AA, and using the DMS(P) as supporting evidence.

Stating clear hypotheses at the beginning of the manuscript, addressing any significant errors mentioned below and splitting Results and Discussion will make for a very strong manuscript that will significantly improve knowledge about AA cycling.

Major comments

Line 109: Only dissolved AA was measured. Please make this clear and consistent with abbreviations for DMSP.

Line 161, 209, 234,241: Riverine/terrestrial runoff is argued to be a critical input of AA into the systems studied but are lacking direct evidence. Are there actual measurements of riverine AA concentrations in Liu 2001 that could be reported ? How do their measurements compare to yours?

Line 173: I only see that DMSPt and Chla coupled (e.g. lowest DMS corresponds to highest Chla). Please edit so as not to overstate trends, and use qualitative statements and tests for significance.

Line 174, 198: Correlations are likely impacted by measuring only dissolved AA, as the majority of AA produced from DMSPd degradation would be expected to be stored intracellularly, whereas the majority of DMS produced would be expected to be found in the dissolved phase. As well, DMS is more diffusive and reactive, and therefore inputs of DMS are likely more complicated than dissolved AA (Tyssebotn et al. 2017). Please consider these comments in the Discussion.

Line 175-177: It is well-established that Chla rarely correlates with sulfur compounds because production is specific to community composition/location (Lana et al. 2011; Galí et al. 2015; McParland and Levine 2019). I suggest authors review comments about these relationships throughout manuscript. Incorporating new parameters (phytoplankton type abundances and bacterial abundances) will better reflect the role of biology in these dynamics.

Line 178-192: Such high AA concentrations in porewater is very interesting, and should be better highlighted…these concentrations are ~an order of magnitude greater than in the water column! I suggest making qualitative comparisons with previously measured AA concentrations and highlighting the significance of these new measurements.

Line 182: Why are the concentrations so different? Location/sediment types? This would be an ideal place to discuss bacterial abundances from previous studies if appropriate.

Line 188: If AA in porewater and bottom water are not significantly correlated, what is the supporting evidence for the statement that the source of high AA in bottom water is porewater?

Line 209-212: Figure 3 as well as associated text are confusing. Are these relationships significant? Are the slopes significantly different than zero? (They do not look so). I'm also confused why AA was normalized to salinity as this is the x-axis? You'll get the same answer. As is, I would remove Figure 3. The relationships do not look significant and do not support conclusions.

Line 226: I'm confused by conclusion that this negative correlation is linked to enhanced lyase activity? If low salinity promotes lyase activity, then we would expect a positive correlation of salinity and DMSPt (i.e. low salinity=more lyase activity=less DMSPt/DMSPp due to cleavage).

Line 234: At the surface, where terrestrial runoff is expected to impact concentrations, the excess does not appear to be 'significant'…(AA at 10m ~60nM, DMSPt at 10m ~55nM, difference =5). Please edit this statement for clarity using quantitative statements and/or justify the use of 'significance' when describing the excess difference in AA and DMSPt.

Line 247: Is there a statistically significant relationship between Chla and DMSPt? Please use qualitative statements, rather than listing the order of concentrations.

Line 255: Please revise this statement…yes DMSP could be a cryoprotectant, but this is most relevant to ice algae and temperatures in freezing conditions.

Line 280: This entire section (3.5) needs statistical tests to support statements. Example: are the rates of DMS production *significantly* lower than rates of DMSPd degradation in summer? (Remember to report the statistical test and p-values in text/methods).

Line 280: Was Chla measured at beginning of experiments? This could better support statements about biomass productivity altering rate measurements (Cho and Azam 1990).

General comment: There is a significant order of magnitude difference between absolute AA concentrations presented here and recently published measurements from the Gulf of Mexico. As

well, uptake rates of AA were are an order of magnitude less than the degradation rates of AA measured here (Tyssebotn et al. 2017). Please acknowledge these previous measurements and describe potential reasons for differences. The AA dynamics presented here by Wu et al. are an exciting contribution to our knowledge of AA and should be compared with all previous work.

General comment: I recommend the authors consider how the measurements of AA dynamics here could help inform a better understanding of the bacterial 'switch' hypothesis for which the environmental drivers of are still debated (Kiene et al. 2000; Slezak et al. 2007; Levine et al. 2012).

Minor comments

Overall the manuscript should be 'cleaned up' in terms of English but also small text errors. Some errors 'overstate' the significance of the statement, but most do not inhibit reading.

Line 29: Please rephrase statement about acid rain. DMS is correlated with the natural acidity of rain (as stated now, implies that DMS is the cause of acid rain).

Line 41: Please rephrase minor producers to 'low producers'.

Line 41: I suggest removing the "For example" part of this sentence as you state all of the well-known high producers (i.e. it is not an example). When describing low producers mention other low producer types (McParland and Levine 2019).

Line 43: this sentence is repeating line 39, please be more concise and edit accordingly

Line 47: AA should be defined here (even though it is properly defined in Abstract)

Line 54: Kinsey and Kieber 2016 is incorrect citation for this statement

Line 55: The use of 'always' here is too strong for the current state of knowledge

Line 58: Alcolombri et al. 2015 is incorrect citation, this paper does not measure anything in situ. Additionally, I would expand these citations as there are so many more studies that have conducted the work described in this sentence besides the two.

Line 80: complicate**d**

Line 86: How was surface sediment sampled? And where? What time of day collected?

Line 91: How was DMS sampled?

Line 94: Was the pre-filtered DMS sample gravity filtered? Please provide a citation for this method. Also, what size GF/F filter was used?

Line 101, 117, 120: Were analytical samples run? (in duplicate, triplicate?)

Line 102: Again, what size GF/F filter was used?

Line 104: How long were the samples allowed to oxidize for?

Line 124: Has this methodology for incubations been performed before? Please cite if so.

Line 126: Why were syringes used for incubation? Were they gas-tight?

Line 131: I don't believe Kiene et al. or Vila-Costa et al. address preferential GB uptake?

Line 132: Please address how rates were calculated? Were regressions/fits statistically significant?

Line 147: Kiene 1996 is incorrect citation, they did not measure AA in their study?

Line 150: Suggestion if you do split into a Results and Discussion section… results for Section 3.1 and 3.2 should be combined for a clearer description of the differences in summer and winter.

Line 151: How are the contours spaced? Center of sea contour looks like 5 ug/L, not 7.07ug/L?

Line 163, 170: Chengshan Cape and Changjiang Estuary not on maps

Line 172: The comment about MS thesis belongs in methods

Line 178-192: I suggest moving results for porewaters to be part of the depth profile results as it seems more relevant to depth distributions, not surface distributions.

Line 185-187: This sentence should be re-written for clarity

Line 198: 'was not correlated with' (remove the word 'any')

Line 203: I think this should read 'main phytoplankton type' ? Species likely changed based on the small/big cell statement following

Line 204: should read 'small diatoms in winter and larger diatoms in summer'

Line 214: As you discuss everything in context of North to South in the preceding text, for clarity I would order these transects in the same way (same for the order in Figures 4,5 and Table 1).

Line 213: If you split Results section, results of Section 3.3 and 3.4 could be combined for clarity.

Line 218: "Concentrations of Chla, AA, DMS" remove this sentence, it should be a part of caption.

Line 219: 'Both Chla and DMS did not displayed…' this sentence does not make sense to me

Line 230: suggested change "…and highest concentrations were observed in…"

Line 239: Correlations are not causation… using the word 'prove' is an overstatement, please edit.

Line 241: I'm confused by this statement. What did Asher et al. 2017 find that indicates the order of average concentrations demonstrates that both DMSPd and DMSPp produce DMS? Please edit.

Line 255-260: DMS(P) correlations with both salinity and temperature may be due to a co-correlation of these abiotic parameters themselves, please use caution in stating these conclusions and incorporate statistical tests appropriately.

Line 258-259: Kiene and Service 1991 looked at DMS production from *dissolved* DMSP, not particulate. I believe you are discussing a correlation of *total DMSP*. Please edit for clarity.

Line 269: This should be "Figs. 4 and 5" I believe?

Line 294: should be "low bacterial abundance" instead of 'poor'

Line 336: what does "and so on" refer to? An unknown source? Please be more specific.

Figure/Table comments

Figure 2: I find the labels of summer/winter and Chla/AA on the plots very helpful but please also label the panels (a,b,c,d) in figure and reference in the caption (consistent with other figures)

Figure 3: As mentioned above, I do not think this figure is necessary and it may be more useful to replace with similar surface plots of DMS and DMSP for summer and winter (like Figure 2).

Figure 4 and 5: Again, reversing the order that the transects are presented to be North to South will make figure clearer. Also the method for interpolating contours should be reported (either in figure captions or methods), and the black dots (I assume sampling points) should be described. Adding temperature for other transects would make for better consistency.

Table 1: Again, I recommend ordering table to be North to South. Caption should better define what 'Surface' refers to (all three sampling sites?) and what depth the transect values reported are.

Table 2: What correlation test was used? Additionally, please address in the methods how temperature and salinity were measured (i.e. CTD profile or was salinity actually measured?).

Table 4: Very minor, but the table would be easier to read if the abbreviation of BS and SYS are added above the transect station names. Also, these experiments were reported to be conducted in duplicate so please report biological errors for rate measurements.

References
Cho, B., and F. Azam. 1990. Biogeochemical significance of bacterial biomass in the ocean's euphotic zone. Mar. Ecol. Prog. Ser. **63**: 253–259. doi:10.3354/meps063253

Galí, M., E. Devred, M. Levasseur, S. J. Royer, and M. Babin. 2015. A remote sensing algorithm for planktonic dimethylsulfoniopropionate (DMSP) and an analysis of global patterns. Remote Sens. Environ. **171**: 171–184. doi:10.1016/j.rse.2015.10.012

Kiene, R. P., L. J. Linn, and J. A. Bruton. 2000. New and important roles for DMSP in marine microbial communities. J. Sea Res. **43**: 209–224.

Lana, a., T. G. Bell, R. Simó, and others. 2011. An updated climatology of surface dimethlysulfide concentrations and emission fluxes in the global ocean. Global Biogeochem. Cycles **25**: n/a-n/a. doi:10.1029/2010GB003850

Levine, N. M., V. a Varaljay, D. a Toole, J. W. H. Dacey, S. C. Doney, and M. A. Moran. 2012. Environmental, biochemical and genetic drivers of DMSP degradation and DMS production in the Sargasso Sea. Environ. Microbiol. **14**: 1210–23. doi:10.1111/j.1462-2920.2012.02700.x

McParland, E. L., and N. M. Levine. 2019. The role of differential DMSP production and community composition in predicting variability of global surface DMSP concentrations. Limnol. Oceanogr. **64**: 757–773. doi:10.1002/lno.11076

Slezak, D., R. P. Kiene, D. a. Toole, R. Simó, and D. J. Kieber. 2007. Effects of solar radiation on the fate of dissolved DMSP and conversion to DMS in seawater. Aquat. Sci. **69**: 377–393. doi:10.1007/s00027-007-0896-z

Tyssebotn, I. M. B., J. D. Kinsey, D. J. Kieber, R. P. Kiene, A. N. Rellinger, and J. Motard-Côté. 2017. Concentrations, biological uptake, and respiration of dissolved acrylate and dimethylsulfoxide in the northern Gulf of Mexico. Limnol. Oceanogr. **62**: 1198–1218. doi:10.1002/lno.10495

---

## Author Comment (AC2) · 3 Sep 2019

Wu et al. measured DMS(P) and AA concentrations across different oceanographic regimes, depths and seasons, rate measurements of DMS(P) and AA degradation and production, and AA concentrations in porewaters. AA is a product of DMSP cleavage and potentially an important carbon source, but little is known about global AA dynamics. I commend the authors for their expansive assessment of AA dynamics in the context of DMS(P) cycling. These measurements reflect an important contribution to knowledge about AA's role in the marine microbial ecosystem. However, the current manuscript requires significant improvements for accuracy and presentation clar-

ity. Specifically: 1. Statistical tests are missing/incomplete throughout the manuscript. Any conclusions deemed significant should be supported by statistics. Overall, results should be made more quantitative.

Reply: According to the reviewer's suggestion, we will add more quantitative descriptions in Results section and statistics to support conclusions deemed significant in the revised manuscript.

2. More biological measurements are necessary to support conclusions. Only Chla concentrations are reported, which is well established to be a poor predictor of DMS(P) concentrations. This is not a focus here as authors have already reported they can add more biological parameters.

Reply: We agree with the reviewer. We will discuss other biological parameters including nutrients, phytoplankton and bacterial data in the revised manuscript.

3. There are a significant number of citation errors in (both missing and incorrect citations) and I highly suggest the authors review their citations fully before resubmitting. Additionally, many conclusions are "overstated", meaning the strength of the wording should be edited.

Reply: Thank you for your suggestion. We will check all citations very carefully and correct the errors. And the strength of the wording will be improved dramatically in the revised manuscript.

4. The clarity of the manuscript would greatly benefit from dividing the Results into Results and Discussion. As it reads now, the results for each section are being explained in pieces but no full story of all the results is tied together.

Reply: According to the reviewer's suggestion, we will divide the previous Results section into Results and Discussion sections in the revised manuscript.

5. Finally, the motivation of the manuscript should be clearer. I fully recognize that these measurements of AA will improve knowledge, but why is it important to fill that

gap? What unknowns do these results answer about AA cycling? Given the expansive AA measurements, this manuscript could test more specific hypotheses. Additionally, for writing clarity, I would recommend focusing the questions towards AA, and using the DMS(P) as supporting evidence. Stating clear hypotheses at the beginning of the manuscript, addressing any significant errors mentioned below and splitting Results and Discussion will make for a very strong manuscript that will significantly improve knowledge about AA cycling.

Reply: Many aspects of DMS and DMSP have been well documented, but the processes affecting AA concentrations in marine waters are poorly known. Furthermore, AA is an important source of carbon to the microbial community. Therefore, it is important to fill the gap. These results indicated other potential sources of AA (e.g. terrestrial inputs from rivers) besides production from DMSPd, determined if temperature was a key controlling factor on AA dynamics through winter and summer comparison, and provided new measurements of AA in porewater. We supposed that changes of phytoplankton and bacteria species and abundance played important roles on AA dynamics and expected these hypotheses could be test in this manuscript. We meant to focus on AA and use the DMS(P) as supporting evidence. We will emphasize this in the revised manuscript. According to the reviewer's suggestion, we will state the above-mentioned hypotheses at the beginning of the manuscript, address significant errors and split into Results and Discussion in the revised manuscript.

Major comments Line 109: Only dissolved AA was measured. Please make this clear and consistent with abbreviations for DMSP.

Reply: Yes, only dissolved AA was measured. We will check the entire manuscript and used the abbreviation AAd for the dissolved AA in the revised manuscript.

Line 161, 209, 234,241: Riverine/terrestrial runoff is argued to be a critical input of AA into the systems studied but are lacking direct evidence. Are there actual measurements of riverine AA concentrations in Liu 2001 that could be reported? How do their

measurements compare to yours?

Reply: Liu (2001) found 90 kinds of organic pollutants including acrylic acid in Yalu River, but the exact concentrations were not reported. We could not compare our results with theirs directly, but we thought it could be a direct evidence for the terrestrial input of AA.

Line 173: I only see that DMSPt and Chla coupled (e.g. lowest DMS corresponds to highest Chla). Please edit so as not to overstate trends, and use qualitative statements and tests for significance.

Reply: We are sorry for confusing you. As horizontal distributions of DMS and DMSP in surface seawater of the BS and the YS has been described by Jin (2016), we did not cite those figures from her MS thesis in our previous manuscript, which made you not see their coupled relationships with Chl a. We will add figures of DMS, DMSPd and DMSPp distributions in surface seawater and describe their relationships using quantitative statements and tests for significance, as indicated below. "DMS and DM-SPd presented positive correlations with Chl a (DMS: r=0.418, n=50, p<0.01; DMSPd: r=0.351, n=50, p<0.05)."

Line 174, 198: Correlations are likely impacted by measuring only dissolved AA, as the majority of AA produced from DMSPd degradation would be expected to be stored intracellularly, whereas the majority of DMS produced would be expected to be found in the dissolved phase. As well, DMS is more diffusive and reactive, and therefore inputs of DMS are likely more complicated than dissolved AA (Tyssebotn et al. 2017). Please consider these comments in the Discussion.

Reply: Thanks for your suggestion. We will add these comments in the Discussion section of revised manuscript.

Line 175-177: It is well-established that Chla rarely correlates with sulfur compounds because production is specific to community composition/location (Lana et al. 2011;

Galí et al. 2015; McParland and Levine 2019). I suggest authors review comments about these relationships throughout manuscript. Incorporating new parameters (phytoplankton type abundances and bacterial abundances) will better reflect the role of biology in these dynamics.

Reply: Thanks for your suggestion. We will review comments about these relationships throughout the manuscript and incorporate phytoplankton type abundances and bacterial abundances to better reflect the role of biology in these dynamics in the revised manuscript.

Line 178-192: Such high AA concentrations in porewater is very interesting, and should be better highlighted. . .these concentrations are ∼an order of magnitude greater than in the water column! I suggest making qualitative comparisons with previously measured AA concentrations and highlighting the significance of these new measurements.

Reply: According to the reviewer's suggestion, we will make quantitative comparisons with previously measured AA concentrations in porewater and highlight the significance of these new measurements in the revised manuscript.

Line 182: Why are the concentrations so different? Location/sediment types? This would be an ideal place to discuss bacterial abundances from previous studies if appropriate.

Reply: Yes, location and sediment types could be the reason for different concentrations. According to the reviewer's suggestion, we will discuss bacterial abundances and compare with previous studies in the revised manuscript.

Line 188: If AA in porewater and bottom water are not significantly correlated, what is the supporting evidence for the statement that the source of high AA in bottom water is porewater?

Reply: We are sorry for the inaccurate statement. Nedwell et al. (1994) reported that DMS could emit from sediments to water column, so we speculated AA could also

emit form porewater to bottom seawater. We will sample vertical cores of sediment to measure the variations of AA concentrations in bottom seawater with time using the method referred in Nedwell et al. (1994) in future cruises. At this time, we will revise that statement as "We speculated AA might emit form porewater to bottom seawater".

Line 209-212: Figure 3 as well as associated text are confusing. Are these relationships significant? Are the slopes significantly different than zero? (They do not look so). I'm also confused why AA was normalized to salinity as this is the x-axis? You'll get the same answer. As is, I would remove Figure 3. The relationships do not look significant and do not support conclusions.

Reply: Thank you for your suggestion, we will remove Figure 3.

Line 226: I'm confused by conclusion that this negative correlation is linked to enhanced lyase activity? If low salinity promotes lyase activity, then we would expect a positive correlation of salinity and DMSPt (i.e. low salinity=more lyase activity=less DMSPt/DMSPp due to cleavage).

Reply: We are sorry for confusing you. We agree with the reviewer. We will revise that sentence as below. "DMSP might be expelled extracellularly in order to reestablish cellular osmotic balance as a response to reduced salinity (Deschaseaux et al., 2014), which might have led to the negative correlation between DMSP and salinity."

Line 234: At the surface, where terrestrial runoff is expected to impact concentrations, the excess does not appear to be 'significant'...(AA at 10m ∼60nM, DMSPt at 10m ∼55nM, difference =5). Please edit this statement for clarity using quantitative statements and/or justify the use of 'significance' when describing the excess difference in AA and DMSPt.

Reply: As Yellow River is the world's largest river in terms of sediment load and flows into the NYS, AA may be absorbed on those sediments and sink to bottom. Therefore, terrestrial runoff may impact AA concentrations along the vertical profiles of transect

B12-17 rather than only at the surface. Along the transect B12-17, the average AA concentration was 34.60 nmol L-1 and more than 2 times of that of DMSPt (15.45 nmol L-1). According to the reviewer's suggestion, we will remove the word 'significant' and state it using quantitative statements, as "the average value of AA was more than 2 times of that of DMSPt".

Line 247: Is there a statistically significant relationship between Chla and DMSPt? Please use qualitative statements, rather than listing the order of concentrations.

Reply: No statistically significant relationship was found between Chl a and DMSPt. We will revise that sentence as below. "This suggests that large amounts of phytoplankton biomass may induce high concentrations of DMSPt."

Line 255: Please revise this statement. . .yes DMSP could be a cryoprotectant, but this is most relevant to ice algae and temperatures in freezing conditions.

Reply: Thank you for your suggestion. We will revise that statement as below. "van Rijssel and Gieskes (2002) also found a negative effect of temperature on the DMSP content per volume."

Line 280: This entire section (3.5) needs statistical tests to support statements. Example: are the rates of DMS production significantly lower than rates of DMSPd degradation in summer? (Remember to report the statistical test and p-values in text/methods).

Reply: Thank you for your suggestion. We will add statistical tests and p-values and edit the strength of the wording for the entire section 3.5.

Line 280: Was Chla measured at beginning of experiments? This could better support statements about biomass productivity altering rate measurements (Cho and Azam 1990).

Reply: We are sorry for not measuring Chl a at beginning of experiments. We have the Chl a data at these stations. It may not have a big difference from the Chl a concentrations at beginning of experiments because the seawater used for experiments were

also sampled from the Niskin bottles. We will discuss the relationships between Chl a and reaction rates in the revised manuscript.

General comment: There is a significant order of magnitude difference between absolute AA concentrations presented here and recently published measurements from the Gulf of Mexico. As well, uptake rates of AA were are an order of magnitude less than the degradation rates of AA measured here (Tyssebotn et al. 2017). Please acknowledge these previous measurements and describe potential reasons for differences. The AA dynamics presented here by Wu et al. are an exciting contribution to our knowledge of AA and should be compared with all previous work.

Reply: Thank you for your suggestion. We will compare the absolute AA concentrations and degradation rates of AA with previous work and explore the reasons for the differences in the revised manuscript.

General comment: I recommend the authors consider how the measurements of AA dynamics here could help inform a better understanding of the bacterial 'switch' hypothesis for which the environmental drivers of are still debated (Kiene et al. 2000; Slezak et al. 2007; Levine et al. 2012).

Reply: Thanks for your suggestion. We will discuss how bacteria species and abundance affect AA dynamics in the revised manuscript.

Minor comments Overall the manuscript should be 'cleaned up' in terms of English but also small text errors. Some errors 'overstate' the significance of the statement, but most do not inhibit reading.

Reply: Thank you for your suggestions. We will polish the entire manuscript and correct text errors and wording errors.

Line 29: Please rephrase statement about acid rain. DMS is correlated with the natural acidity of rain (as stated now, implies that DMS is the cause of acid rain).

Reply: Thanks for your suggestion. We will revise this sentence as "DMS is correlated

with the natural acidity of rain."

Line 41: Please rephrase minor producers to 'low producers'.

Reply: Thanks for your suggestion. We will revise 'minor producers' to 'low producers'.

Line 41: I suggest removing the "For example" part of this sentence as you state all of the well-known high producers (i.e. it is not an example). When describing low producers mention other low producer types (McParland and Levine 2019).

Reply: According to the reviewer's suggestion, we will remove the sentence of high producers and describe other low producer types, as indicated below. "DMSP distributions are also controlled by phytoplankton species, among which diatoms, flagellates, Prochlorophytes and cyanobacteria are low minor producers of DMSP (McParland and Levine 2019)."

Line 43: this sentence is repeating line 39, please be more concise and edit accordingly

Reply: According to the reviewer's suggestion, we will delete the sentence in line 43 and rephrase the sentence in line 39, as indicated below. "As an antioxidant, a cryoprotectant, and an osmolyte in marine phytoplankton, the production of DMSP is influenced by environmental parameters such as salinity (Stefels, 2000), temperature (Kirst et al., 1991), and oxidative stress (Sunda et al., 2002)."

Line 47: AA should be defined here (even though it is properly defined in Abstract)

Reply: Yes. We defined AA as the abbreviation of acrylic acid in line 46 when it was first mentioned in text.

Line 54: Kinsey and Kieber 2016 is incorrect citation for this statement

Reply: We are sorry for the incorrect citation. We will cite another reference of Noordkamp et al., 2000 for this statement.

Line 55: The use of 'always' here is too strong for the current state of knowledge

Reply: Thank you for your suggestion, we will remove 'always' in the revised manuscript.

Line 58: Alcolombri et al. 2015 is incorrect citation, this paper does not measure anything in situ. Additionally, I would expand these citations as there are so many more studies that have conducted the work described in this sentence besides the two.

Reply: We are sorry for the incorrect citation. We will remove that citation and add others including "Lana et al., 2011; Levine et al., 2012; Tyssebotn et al., 2017".

Line 80: complicated

Reply: Thanks for your correction. We will revise 'complicate' to 'complicated'.

Line 86: How was surface sediment sampled? And where? What time of day collected?

Reply: Sediments were collected using a stainless-steel box-corer and sub-sampled to a depth of ca. 3 cm. They were sampled at 12 stations shown in Table 1 during summer cruise. We will add the sampling method of surface sediment in the revised manuscript and sampling time in revised Table 1, as indicated below. "Sediments were collected using a stainless-steel box-corer and sub-sampled to a depth of ca. 3 cm at 12 stations shown in Table 1 during summer cruise."

Line 91: How was DMS sampled?

Reply: DMS sampling was conducted as soon as the Niskin bottles were on deck. 250 mL brown glass bottle were rinsed and filled to the top to eliminate any headspace in an effort to minimize partitioning into the gas phase. A 2 mL aliquot of seawater sample was directly extracted from the 250 mL brown glass bottle using a 2 mL glass syringe and injected into a glass bubbling chamber. We will add these sentences in the revised manuscript.

Line 94: Was the pre-filtered DMS sample gravity filtered? Please provide a citation for

this method. Also, what size GF/F filter was used?

Reply: We did not filter seawater through GF/F filter before analyzing. We will delete that part from DMS analytical procedures.

Line 101, 117, 120: Were analytical samples run? (in duplicate, triplicate?)

Reply: Analytical samples were run in duplicate. We will add this sentence at the end of Section 2.3 in the revised manuscript.

Line 102: Again, what size GF/F filter was used?

Reply: 47 mm GF/F filter was used here. We will add the size in the revised manuscript.

Line 104: How long were the samples allowed to oxidize for?

Reply: The samples were allowed to oxidize for 2 days. We will add a sentence in the revised manuscript as indicated below. "To fully oxidize pre-existing gaseous DMS, the DMSPt and DMSPd samples were incubated in the dark at room temperature for 2 days."

Line 124: Has this methodology for incubations been performed before? Please cite if so.

Reply: Yes. GBT inhibition method for DMSPd degradation was performed by Kiene and Gerard (1995). Methods for photochemical and microbial degradation of AA were performed by Wu et al. (2015). We will add these citations in the revised manuscript.

Line 126: Why were syringes used for incubation? Were they gas-tight?

Reply: Yes. These syringes were gas-tight. It was convenient to collect samples at 0, 3, and 6 h if using syringes. We could just push the plunger to let seawater flow out. Meanwhile, it kept the rest seawater in syringes isolated from air.

Line 131: I don't believe Kiene et al. or Vila-Costa et al. address preferential GB uptake?

Reply: We are sorry for misunderstanding these articles. We will revise that sentence and cite another reference, as indicated below. "and acts as a competitive inhibitor of DMSP (Kiene et al., 1998)."

Line 132: Please address how rates were calculated? Were regressions/fits statistically significant?

Reply: According to the reviewer's suggestion, we will add description about rates calculation, as indicated below. "Linear regression equations were fit to the DMSPd, DMS and AAd time course data and the apparent rates were estimated as the differences between the slopes of samples with and without GBT." Regressions at most stations were statistically significant.

Line 147: Kiene 1996 is incorrect citation, they did not measure AA in their study?

Reply: No, Kiene (1996) did not measure AA in his study. He determined the kinetics of DMSP(d) degradation by running one with spike additions of DMSP and the other one without additions as control. We thought this method could be applied to AA degradation, so we cited this article. We will remove this citation as the reviewer suggested.

Line 150: Suggestion if you do split into a Results and Discussion section. . . results for Section 3.1 and 3.2 should be combined for a clearer description of the differences in summer and winter.

Reply: According to the reviewer's suggestion, we will split into Results and Discussion sections and combine results for Section 3.1 and 3.2.

Line 151: How are the contours spaced? Center of sea contour looks like 5 ug/L, not 7.07ug/L?

Reply: Kriging method was used for interpolating contours. These circles inside the contour of 5 $\mu$g L-1 were too small to be marked as their real concentrations. As we measured, the center point was station B61 with the Chl a concentration of 7.07 $\mu$g L-1.

Line 163, 170: Chengshan Cape and Changjiang Estuary not on maps

Reply: According to the reviewer's suggestion, we will add Chengshan Cape and Changjiang Estuary on maps.

Line 172: The comment about MS thesis belongs in methods

Reply: According to the reviewer's suggestion, we will move the comment about MS thesis to Material and methods section.

Line 178-192: I suggest moving results for porewaters to be part of the depth profile results as it seems more relevant to depth distributions, not surface distributions.

Reply: According to the reviewer's suggestion, we will move results for porewaters to be part of the depth profile results.

Line 185-187: This sentence should be re-written for clarity

Reply: According to the reviewer's suggestion, we will re-write this sentence, as indicated below. "The large amounts of intracellular DMSP could be cleaved to AAd by DddY, which is as the only known periplasmic DMSP lyase and widely present in $\beta$-, $\gamma$-, $\delta$- and $\varepsilon$-proteobacteria which are the dominant bacteria communities in the surface sediments of the BS and the YS (Li et al., 2017;Liu et al., 2015a;Xie et al., 2017)."

Line 198: 'was not correlated with' (remove the word 'any')

Reply: According to the reviewer's suggestion, we will remove the word 'any'.

Line 203: I think this should read 'main phytoplankton type'? Species likely changed based on the small/big cell statement following

Reply: Yes, we will revise 'main phytoplankton species' to 'main phytoplankton types'.

Line 204: should read 'small diatoms in winter and larger diatoms in summer'

Reply: According to the reviewer's suggestion, we will revise to 'small diatoms in winter and larger diatoms in summer'.

Line 214: As you discuss everything in context of North to South in the preceding text, for clarity I would order these transects in the same way (same for the order in Figures 4,5 and Table 1).

Reply: Thanks for your suggestion, we will order these transects North to South in text, Figures 4, 5 and Table 1.

Line 213: If you split Results section, results of Section 3.3 and 3.4 could be combined for clarity.

Reply: According to the reviewer's suggestion, we will split into Results and Discussion sections and combine results of Section 3.3 and 3.4 in the revised manuscript.

Line 218: "Concentrations of Chla, AA, DMS" remove this sentence, it should be a part of caption.

Reply: According to the reviewer's suggestion, we will remove that sentence in the revised manuscript.

Line 219: 'Both Chla and DMS did not displayed. . .' this sentence does not make sense to me

Reply: We are sorry for confusing you. We will revise that sentence to 'Neither Chl a nor DMS displayed. . .'.

Line 230: suggested change ". . .and highest concentrations were observed in. . ."

Reply: Thank you for your suggestion. We will change that sentence to ". . .and highest concentrations were observed in. . .".

Line 239: Correlations are not causation. . . using the word 'prove' is an overstatement, please edit.

Reply: Thank you for your suggestion. We will change the word 'prove' to 'indicate'.

Line 241: I'm confused by this statement. What did Asher et al. 2017 find that indicates the order of average concentrations demonstrates that both DMSPd and DMSPp produce DMS? Please edit.

Reply: We are sorry for the incorrect citation and statement. We will revise this sentence, as indicated below. "The higher values of DMS than DMSPd might be produced through the intra-cellular cleavage of phytoplankton DMSPp catalyzed by the enzyme DMSP lyase and the photochemical and biological reduction of DMSO to DMS, while the higher values of AAd than DMSPt indicated that there were terrestrial sources of AAd besides the contribution from in situ DMSP degradation along the three transects."

Line 255-260: DMS(P) correlations with both salinity and temperature may be due to a cocorrelation of these abiotic parameters themselves, please use caution in stating these conclusions and incorporate statistical tests appropriately.

Reply: We agree with the reviewer. We will add this comment in the revised manuscript.

Line 258-259: Kiene and Service 1991 looked at DMS production from dissolved DMSP, not particulate. I believe you are discussing a correlation of total DMSP. Please edit for clarity.

Reply: Thanks for your suggestion. We will revise 'DMSP' to 'DMSPd' for clarity.

Line 269: This should be "Figs. 4 and 5" I believe?

Reply: We are sorry for the typos. We will correct "Figs. 3 and 4" to "Figs. 4 and 5" in the revised manuscript.

Line 294: should be "low bacterial abundance" instead of 'poor'

Reply: Thank you for your suggestion. We will revise it to "low bacterial abundance".

Line 336: what does "and so on" refer to? An unknown source? Please be more specific.

Reply: According to the reviewer's suggestion, we will revise "and so on" to "other

unknown sources".

Figure/Table comments Figure 2: I find the labels of summer/winter and Chla/AA on the plots very helpful but please also label the panels (a,b,c,d) in figure and reference in the caption (consistent with other figures)

Reply: According to the reviewer's suggestion, we will label the panels (a, b, c, d) in figure and refer them in the caption.

Figure 3: As mentioned above, I do not think this figure is necessary and it may be more useful to replace with similar surface plots of DMS and DMSP for summer and winter (like Figure 2).

Reply: According to the reviewer's suggestion, we will remove that figure and replaced with surface plots of DMS and DMSP for summer and winter shown in Jin (2016) and Sun (2017).

Figure 4 and 5: Again, reversing the order that the transects are presented to be North to South will make figure clearer. Also the method for interpolating contours should be reported (either in figure captions or methods), and the black dots (I assume sampling points) should be described. Adding temperature for other transects would make for better consistency.

Reply: Thank you for your suggestions. We will order the transects North to South, report the kriging method for interpolating contours in figure captions, describe the black dots as sampling points, and add temperature for other transects.

Table 1: Again, I recommend ordering table to be North to South. Caption should better define what 'Surface' refers to (all three sampling sites?) and what depth the transect values reported are.

Reply: We will order the transects North to South. 'Surface' refers to "surface seawater of the whole study area (the BS and the YS)". The transect values are the average of the whole vertical profile of each transect. We will define these in Table 1 caption in the

revised manuscript.

Table 2: What correlation test was used? Additionally, please address in the methods how temperature and salinity were measured (i.e. CTD profile or was salinity actually measured?).

Reply: Pearson correlation test was used here. We will add this sentence in figure caption and address CTD profiles of temperature and salinity in Material and methods section.

Table 4: Very minor, but the table would be easier to read if the abbreviation of BS and SYS are added above the transect station names. Also, these experiments were reported to be conducted in duplicate so please report biological errors for rate measurements.

Reply: Thank you for your suggestions. We will add the abbreviation of BS and SYS above the transect station names and report biological errors for rate measurements in revised Table 4.

References Cho, B., and F. Azam. 1990. Biogeochemical significance of bacterial biomass in the ocean's euphotic zone. Mar. Ecol. Prog. Ser. 63: 253–259. doi:10.3354/meps063253 Galí, M., E. Devred, M. Levasseur, S. J. Royer, and M. Babin. 2015. A remote sensing algorithm for planktonic dimethylsulfoniopropionate (DMSP) and an analysis of global patterns. Remote Sens. Environ. 171: 171–184. doi:10.1016/j.rse.2015.10.012 Kiene, R. P., L. J. Linn, and J. A. Bruton. 2000. New and important roles for DMSP in marine microbial communities. J. Sea Res. 43: 209–224. Lana, a., T. G. Bell, R. Simó, and others. 2011. An updated climatology of surface dimethlysulfide concentrations and emission fluxes in the global ocean. Global Biogeochem. Cycles 25: n/a-n/a. doi:10.1029/2010GB003850 Levine, N. M., V. a Varaljay, D. a Toole, J. W. H. Dacey, S. C. Doney, and M. A. Moran. 2012. Environmental, biochemical and genetic drivers of DMSP degradation and DMS production in the Sargasso Sea. Environ. Microbiol. 14: 1210–23. doi:10.1111/j.1462- 2920.2012.02700.x

McParland, E. L., and N. M. Levine. 2019. The role of differential DMSP production and community composition in predicting variability of global surface DMSP concentrations. Limnol. Oceanogr. 64: 757–773. doi:10.1002/lno.11076 Slezak, D., R. P. Kiene, D. a. Toole, R. Simó, and D. J. Kieber. 2007. Effects of solar radiation on the fate of dissolved DMSP and conversion to DMS in seawater. Aquat. Sci. 69: 377–393. doi:10.1007/s00027-007-0896-z Tyssebotn, I. M. B., J. D. Kinsey, D. J. Kieber, R. P. Kiene, A. N. Rellinger, and J. Motard-Côté. 2017. Concentrations, biological uptake, and respiration of dissolved acrylate and dimethylsulfoxide in the northern Gulf of Mexico. Limnol. Oceanogr. 62: 1198–1218. doi:10.1002/lno.10495

Reply: Thank you for listing these references. We will add them in the revised manuscript.

---

## Author Response (AR1)

Dear Prof. Dai,

Thank you for your kind consideration and constructive comments for our manuscript entitled 'Acrylic acid and related dimethylated sulfur compounds in the Bohai and Yellow Seas during summer and winter'. We are grateful to the anonymous reviewers for their constructive suggestions, which is of great help to improve the manuscript. Please find our final responses (in blue) and changes (in red) to all comments (in black) in this document.

**Response to reviewer #1**

Comments from reviewer #1 are in black while our responses are in blue and changes in the manuscript are in red.

Review of BG-2019-172 by Wu et al. This paper describes the DMS/P and AA surface ocean cycling in the Bohai and Yellow Seas during winter and summer. The authors also measured depth profiles and porewater concentrations, as well as performed incubation experiments to derive production/degradation rates. This paper contains valuable data, but only a small amount of new science. By now, the community has a generally good understanding of DMS dynamics and the controlling factors. We know that phytoplankton, bacteria, and environmental parameters influence DMS/P cycling (and related compounds). Nonetheless, it appears that the authors did not measure phytoplankton or bacterial parameters. They attempt to explain processes without measuring the parameters involved. This paper is generally more suited to a journal like ESSD.

Thanks for all the constructive comments and helpful suggestions to improve this manuscript.
We found phytoplankton and bacterial data of these two cruises in two published papers (Zhang 2018; Liang et al., 2019). We have discussed how these parameters influenced AA and DMS/P cycling in revised manuscript. In addition, our study proved other sources of AA (e.g. terrestrial inputs from rivers and production from DMSPp) in surface seawater through on-deck incubation experiments. Although some observations and studies on the distributions of DMS and DMSP in the Bohai Sea and the Yellow Sea have been conducted (Yang et al., 2014; 2015; Li et al., 2016), the study aiming at winter has not been reported as well as the relationship between AA and DMS/P, which could reflect if temperature was a key controlling factor on biogenic sulfur cycling. Furthermore, our study was the first time to collect AA samples in porewater in Chinese marginal seas, although more work needs to be done to further understand the source and fate of AA in marine sediments. We have strengthened these particularities in Section 1 of the revised manuscript, as indicated below.

"Many aspects of DMS and DMSP including spatio-temporal distributions, degradation, sea-to-air fluxes, and particle size fractionation have been well documented (Lana et al., 2011; Levine et al., 2012; Yang et al., 2014; Espinosa et al., 2015; Tyssebotn et al., 2017). Up to date, however, the biogeochemistry of AA itself in the oceans and the roles of AA in the marine sulfur cycle and the microbial community has received only limited attention. Tan et al. (2017) and Wu et al. (2017) reported spatial distributions of AA in the Changjiang Estuary and the East China Sea. Liu et al. (2016) investigated the spatial and diurnal variations of AA in the Bohai Sea (BS) and Yellow Sea (YS) during autumn and measured the apparent production rates of AA through DMSP degradation by incubations. However, seasonal variations, source and removal of AA, and the key factors controlling these processes still remain unclear, and thus further studies are needed to better understand the biogeochemical cycle of sulfur in the oceans. In this study, we investigated horizontal and vertical distributions of dissolved AA (AAd) and related dimethylated sulfur compounds in the BS and YS in different seasons (summer and winter) to determine if temperature, phytoplankton and bacteria

species and abundance were key controlling factors on AA dynamics. In addition, it was the first time to collect AAd samples in porewater of surface sediment during summer in the BS and YS. We also examined the degradation of dissolved DMSP (DMSPd) and AAd simultaneously through on-deck incubations during summer and winter to understand production and consumption mechanisms of AA, DMS, and DMSP, to explore the influencing factors (i.e. changes of bacteria species and abundance) of microbial degradation, and to indicate other potential sources of AA. This study is expected to provide insightful information on sulfur cycling from the view of AA in the marginal seas." (L55-71)

Specific comments:
1. The English throughout the entire manuscript needs to be slightly revised. Overall, the language is fine, but there are still many mistakes.

Thanks for your suggestions. We have checked the entire manuscript to polish it and correct mistakes.

2. Section 2.3 Were there particulate measurements of anything (no filtering to measure total DMS/P, etc.)? Did you measure duplicates or triplicates? How exactly was precision and the limit of detection determined?

We measured total DMSP (DMSPt, no filtering) and dissolved DMSP (DMSPd, filtering with 0.7 μm GF/F). We did not measure particulate DMSP (DMSPp) directly, but DMSPp can be calculated using DMSPt minus DMSPd.
Duplicates were measured.
Because DMSP is converted to DMS and then measured, the precision and the limit of detection for DMSP are same as those for DMS, namely, the analytical precision is generally better than 10% and the detection limit is 0.4 nmol $L^{-1}$,
We have added these descriptions to Section 2.3, as indicated below.
    "A 10 mL aliquot of seawater without filtering was used for total DMSP (DMSPt) analysis." (L110)
    "This method gave the same precision and detection limit for DMSP as DMS." (L116-117)
    "Analytical samples for DMS, DMSPd, DMSPt, AAd, Chl *a*, and nutrients were run in duplicate." (L135-136)

3. Section 2.3 Were nutrients measured? Phytoplankton pigments or flow cytometry?

Nutrients were measured. The Utermöhl method was used for phytoplankton counting described in Zhang's thesis (2018). We have added analytical procedures in Section 2.3, as indicated below.
    "In addition, the concentrations of nutrients (including $PO_4^{3-}$, $NO_3^-$, $NO_2^-$, $NH_4^+$, and $SiO_3^{2-}$) were analyzed using a nutrient automatic analyzer (Auto Analyzer 3, SEAL Analytical, USA). Phytoplankton data recorded by Utermöhl method and bacteria data measured by qPCR were collected from Zhang (2018) and Liang et al. (2019), respectively." (L132-135)

4. Section 3.5 Were bacterial parameters measured? Why not? Did you see evidence of first order rates? Did you discover something new with the incubation experiments?

We are sorry for not measuring the bacterial parameters, but we found a published paper (Liang et

al., 2019) discussing the bacterial parameters of the same cruises. We have used this data to support our experiments and cited this paper in revised manuscript.

In our published paper (Wu et al., 2015) which studied the acrylic acid (AA) degradation in details, we did incubation experiments for 8 h and sampled every 2 h. It was found that AA degraded quickly in first 2 h, the degradation rates reduced gradually, and the loss curves fit the first-order equation. Kiene (1996) also demonstrated that apparent first order rate constants ($k$) for the loss of DMSPd were estimated by plotting the natural log of the DMSPd concentration vs time.

Besides the DMSPd degradation experiments, we carried out the AA biological and photochemical degradation experiments simultaneously. We found the total consumption (biological + photochemical) rates of AA were always higher than the production rates of AA from DMSPd at different stations during these two cruises, which provided evidence for other sources of AA in this study area.

The following references are added.

Liang, J., Liu, J., Wang, X., Lin, H., Liu, J., Zhou, S., Sun, H., and Zhang, X.-H.: Spatiotemporal dynamics of free-living and particle-associated Vibrio communities in the northern Chinese marginal seas, Appl. Environ. Microbiol., 85, e00217-00219, 2019.

Zhang, D.: The study of phytoplankton and biosilicon in the Yellow Sea and the Bohai Sea (in Chinese with English abstract), MS thesis, Tianjin University of Science & Technology, Tianjin, China, 2018.

**Response to reviewer #2**

Comments from reviewer #2 are in black while our responses are in blue and changes in the manuscript are in red.

Wu et al. measured DMS(P) and AA concentrations across different oceanographic regimes, depths and seasons, rate measurements of DMS(P) and AA degradation and production, and AA concentrations in porewaters. AA is a product of DMSP cleavage and potentially an important carbon source, but little is known about global AA dynamics. I commend the authors for their expansive assessment of AA dynamics in the context of DMS(P) cycling. *These measurements reflect an important contribution to knowledge about AA's role in the marine microbial ecosystem.* However, the current manuscript requires significant improvements for accuracy and presentation clarity.

We have carefully considered the reviewer' comments and suggestions and conducted the revision seriously. We are very grateful to the reviewer for all the constructive comments and helpful suggestions to improve this manuscript.

Specifically:
1. Statistical tests are missing/incomplete throughout the manuscript. Any conclusions deemed significant should be supported by statistics. Overall, results should be made more quantitative.

According to the reviewer's suggestion, we have added more quantitative descriptions and statistics to support conclusions deemed significant in the revised manuscript.

2. More biological measurements are necessary to support conclusions. Only Chla concentrations are reported, which is well established to be a poor predictor of DMS(P) concentrations. This is not a focus here as authors have already reported they can add more biological parameters.

We agree with the reviewer. We have discussed the effects of other biological parameters including nutrients, phytoplankton and bacterial abundance on AA and DMS(P) distributions and dynamics in the revised manuscript.

3. There are a significant number of citation errors in (both missing and incorrect citations) and I highly suggest the authors review their citations fully before resubmitting. Additionally, many conclusions are "overstated", meaning the strength of the wording should be edited.

Thank you for your suggestion. We have checked all citations very carefully and corrected the errors. And the strength of the wording has been improved dramatically in the revised manuscript.

4. The clarity of the manuscript would greatly benefit from dividing the Results into Results and Discussion. As it reads now, the results for each section are being explained in pieces but no full story of all the results is tied together.

According to the reviewer's suggestion, we have divided the previous Results section into Results and Discussion sections in the revised manuscript.

5. Finally, the motivation of the manuscript should be clearer. I fully recognize that these

measurements of AA will improve knowledge, but why is it important to fill that gap? What unknowns do these results answer about AA cycling? Given the expansive AA measurements, this manuscript could test more specific hypotheses. Additionally, for writing clarity, I would recommend focusing the questions towards AA, and using the DMS(P) as supporting evidence. Stating clear hypotheses at the beginning of the manuscript, addressing any significant errors mentioned below and splitting Results and Discussion will make for a very strong manuscript that will significantly improve knowledge about AA cycling.

Many aspects of DMS and DMSP have been well documented, but the processes affecting AA concentrations in marine waters are poorly known. Furthermore, AA is an important source of carbon to the microbial community and high concentration of AA can inhibit bacterial activity, which is very important for studying marine sediment ecosystem. Therefore, it is important to fill the gap.

These results indicated other potential sources of AA (e.g. terrestrial inputs from rivers) besides production from DMSPd, determined if temperature was a key controlling factor on AA dynamics through winter and summer comparison, and provided new measurements of AA in porewater.

We supposed that changes of phytoplankton and bacteria species and abundance played important roles on AA dynamics and expected these hypotheses could be test in this manuscript.

We meant to focus on AA and use the DMS(P) as supporting evidence. We have emphasized this in the revised manuscript.

According to the reviewer's suggestion, we have stated the above-mentioned hypotheses in Section 1 as indicated below, addressed significant errors and split into Results and Discussion in the revised manuscript.

"Many aspects of DMS and DMSP including spatio-temporal distributions, degradation, sea-to-air fluxes, and particle size fractionation have been well documented (Lana et al., 2011; Levine et al., 2012; Yang et al., 2014; Espinosa et al., 2015; Tyssebotn et al., 2017). Up to date, however, the biogeochemistry of AA itself in the oceans and the roles of AA in the marine sulfur cycle and the microbial community has received only limited attention. Tan et al. (2017) and Wu et al. (2017) reported spatial distributions of AA in the Changjiang Estuary and the East China Sea. Liu et al. (2016) investigated the spatial and diurnal variations of AA in the Bohai Sea (BS) and Yellow Sea (YS) during autumn and measured the apparent production rates of AA through DMSP degradation by incubations. However, seasonal variations, source and removal of AA, and the key factors controlling these processes still remain unclear, and thus further studies are needed to better understand the biogeochemical cycle of sulfur in the oceans. In this study, we investigated horizontal and vertical distributions of dissolved AA (AAd) and related dimethylated sulfur compounds in the BS and YS in different seasons (summer and winter) to determine if temperature, phytoplankton and bacteria species and abundance were key controlling factors on AA dynamics. In addition, it was the first time to collect AAd samples in porewater of surface sediment during summer in the BS and YS. We also examined the degradation of dissolved DMSP (DMSPd) and AAd simultaneously through on-deck incubations during summer and winter to understand production and consumption mechanisms of AA, DMS, and DMSP, to explore the influencing factors (i.e. changes of bacteria species and abundance) of microbial degradation, and to indicate other potential sources of AA. This study is expected to provide insightful information on sulfur cycling from the view of AA in the marginal seas." (L55-71)

Major comments

Line 109: Only dissolved AA was measured. Please make this clear and consistent with abbreviations

for DMSP.

Yes, only dissolved AA was measured. We have checked the entire manuscript and used the abbreviation AAd for the dissolved AA in the revised manuscript.

Line 161, 209, 234,241: Riverine/terrestrial runoff is argued to be a critical input of AA into the systems studied but are lacking direct evidence. Are there actual measurements of riverine AA concentrations in Liu 2001 that could be reported? How do their measurements compare to yours?

Liu (2001) found 90 kinds of organic pollutants including acrylic acid in Yalu River, but the exact concentrations were not reported. We could not compare our results with theirs directly, but we thought it could be a direct evidence for the terrestrial input of AA.

Line 173: I only see that DMSPt and Chla coupled (e.g. lowest DMS corresponds to highest Chla). Please edit so as not to overstate trends, and use qualitative statements and tests for significance.

We are sorry for confusing you. As horizontal distributions of DMS and DMSP in surface seawater of the BS and the YS has been described by Jin (2016) and Sun (2017), we did not cite those figures from their MS theses in our previous manuscript, which made you not see their coupled relationships with Chl *a*. We have added figures of DMS, DMSPd and DMSPp distributions in surface seawater during summer and winter and described their relationships using quantitative statements and tests for significance, as indicated below.

"Jin (2016) and Sun (2017) found significant positive correlations between DMS(P) and Chl *a* during summer (DMS: $r = 0.418$, $n = 50$, $p < 0.01$; DMSPd: $r = 0.351$, $n = 50$, $p < 0.05$) and winter (DMS: $r = 0.629$, $p < 0.01$; DMSPp: $r = 0.527$, $p < 0.01$), respectively." (L312-314)

Line 174, 198: Correlations are likely impacted by measuring only dissolved AA, as the majority of AA produced from DMSPd degradation would be expected to be stored intracellularly, whereas the majority of DMS produced would be expected to be found in the dissolved phase. As well, DMS is more diffusive and reactive, and therefore inputs of DMS are likely more complicated than dissolved AA (Tyssebotn et al. 2017). Please consider these comments in the Discussion.

Thanks for your suggestion. We have added these comments in the Discussion section of revised manuscript, as indicated below.

"However, AAd showed no correlations with Chl *a*, nutrients, DMS, and DMSP in the whole study area during summer and winter, which were likely impacted by measuring only dissolved AA. The majority of AA produced from DMSPd degradation would be expected to be stored intracellularly (Kinsey et al., 2016; Tyssebotn et al., 2017), whereas the majority of DMS produced would be expected to be found in the dissolved phase (Spiese et al., 2016)." (L316-319)

Line 175-177: It is well-established that Chla rarely correlates with sulfur compounds because production is specific to community composition/location (Lana et al. 2011; Galí et al. 2015; McParland and Levine 2019). I suggest authors review comments about these relationships throughout manuscript. Incorporating new parameters (phytoplankton type abundances and bacterial abundances) will better reflect the role of biology in these dynamics.

Thanks for your suggestion. We have reviewed comments about these relationships throughout the manuscript and incorporated phytoplankton type abundances and bacterial abundances to better reflect the role of biology in these dynamics in the revised manuscript.

Line 178-192: Such high AA concentrations in porewater is very interesting, and should be better highlighted…these concentrations are ~an order of magnitude greater than in the water column! I suggest making qualitative comparisons with previously measured AA concentrations and highlighting the significance of these new measurements.

According to the reviewer's suggestion, we have made quantitative comparisons with previously measured AA concentrations in porewater and highlighted the significance of these new measurements in the revised manuscript, as indicated below.

"The AAd concentrations in porewater in our study were much higher than those (50-60 nmol $L^{-1}$) in Gulf of Mexico reported by Vairavamurthy et al. (1986). The differences might be owing to the differences in sampling, analytical methods and locations. In their study, sediment porewater was obtained by centrifugation of thawed samples that had been kept deep-frozen and they measured only two samples using electron capture gas chromatography, whereas we collected porewater via Rhizon soil moisture samplers connecting to vacuum tubes and analysed samples using a high performance liquid chromatograph. The pressure in vacuum tube might cause cell break in sediments and thus release more AAd in porewater. Moreover, the bacteria abundance and species in the sediments of the BS and YS in 2015 might be different from those in Gulf of Mexico in 1986. Wang (2015) reported δ- and γ-proteobacteria were the dominant taxa in the sediments of the BS and YS, proportion ranging between 24%-70%. Meanwhile, DddY, which is the only known periplasmic DMSP lyase (Li et al., 2017), is widely present in δ- and γ-proteobacteria and can cleave the large amounts of intracellular DMSP (mmol $L^{-1}$ levels) concentrated by DMSP catabolizing bacteria (Wang et al., 2017). Therefore, all those factors led to high AAd concentrations in porewater of surface sediments.

Slezak et al. (1994) discovered that bacterial activity was retarded at AA concentrations > 10 μmol $L^{-1}$ in long-term incubations of seawater cultures (24 to 110 h). Therefore, AAd in porewater might reduce bacterial metabolism and thus impact the microbial community in sediments, which is very important for studying marine sediment ecosystem." (L368-382)

Line 182: Why are the concentrations so different? Location/sediment types? This would be an ideal place to discuss bacterial abundances from previous studies if appropriate.

Yes, location and sediment types could be the reason for different concentrations. According to the reviewer's suggestion, we have discussed bacterial abundances in the revised manuscript, as indicated below.

"Moreover, the bacteria abundance and species in the sediments of the BS and YS in 2015 might be different from those in Gulf of Mexico in 1986. Wang (2015) reported δ- and γ-proteobacteria were the dominant taxa in the sediments of the BS and YS, proportion ranging between 24%-70%. Meanwhile, DddY, which is the only known periplasmic DMSP lyase (Li et al., 2017), is widely present in δ- and γ-proteobacteria and can cleave the large amounts of intracellular DMSP (mmol $L^{-1}$ levels) concentrated by DMSP catabolizing bacteria (Wang et al., 2017)." (L374-378)

Line 188: If AA in porewater and bottom water are not significantly correlated, what is the supporting evidence for the statement that the source of high AA in bottom water is porewater?

We are sorry for the inaccurate statement. Nedwell et al. (1994) reported that DMS could emit from sediments to water column, so we speculated AA could also emit form porewater to bottom seawater. We will sample vertical cores of sediment to measure the variations of AA concentrations in bottom seawater with time using the method referred in Nedwell et al. (1994) in future cruises. At this time, we have revised that statement as "We speculated AA might emit form porewater to bottom seawater". (L383)

Line 209-212: Figure 3 as well as associated text are confusing. Are these relationships significant? Are the slopes significantly different than zero? (They do not look so). I'm also confused why AA was normalized to salinity as this is the x-axis? You'll get the same answer. As is, I would remove Figure 3. The relationships do not look significant and do not support conclusions.

Thank you for your suggestion, we have removed Figure 3.

Line 226: I'm confused by conclusion that this negative correlation is linked to enhanced lyase activity? If low salinity promotes lyase activity, then we would expect a positive correlation of salinity and DMSPt (i.e. low salinity=more lyase activity=less DMSPt/DMSPp due to cleavage).

We agree with the reviewer. We have revised that sentence as below.
    "DMSP showed positive correlations with temperature and negative correlations with salinity along the three transects during summer, while DMS and DMSP presented negative correlations with temperature and salinity during winter, which might be due to a co-correlation of these abiotic parameters themselves." (L333-336)

Line 234: At the surface, where terrestrial runoff is expected to impact concentrations, the excess does not appear to be 'significant'…(AA at 10m ~60nM, DMSPt at 10m ~55nM, difference =5). Please edit this statement for clarity using quantitative statements and/or justify the use of 'significance' when describing the excess difference in AA and DMSPt.

As the Yellow River is the world's largest river in terms of sediment load and flows into the NYS and the depth of transect B12-17 is less than 70 m, AA may be absorbed on those sediments and sink to bottom. Therefore, terrestrial runoff may impact AA concentrations along the vertical profiles of transect B12-17 rather than only at the surface. Along the transect B12-17, the average AA concentration was 34.60 nmol L$^{-1}$ and more than 2 times of that of DMSPt (15.45 nmol L$^{-1}$). According to the reviewer's suggestion, we have removed the word 'significant' and state it using quantitative statements, as "the average value of AA was more than 2 times of that of DMSPt". (L205)

Line 247: Is there a statistically significant relationship between Chla and DMSPt? Please use qualitative statements, rather than listing the order of concentrations.

No statistically significant relationship was found between Chl *a* and DMSPt. We have revised that sentence as below.
    "This suggested that large amounts of phytoplankton biomass might induce high concentrations of DMSPt." (L351-352)

Line 255: Please revise this statement…yes DMSP could be a cryoprotectant, but this is most relevant to ice algae and temperatures in freezing conditions.

Thank you for your suggestion. We have deleted that statement.

Line 280: This entire section (3.5) needs statistical tests to support statements. Example: are the rates of DMS production *significantly* lower than rates of DMSPd degradation in summer? (Remember to report the statistical test and p-values in text/methods).

Thank you for your suggestion. We have added statistical tests and p-values and edit the strength of the wording for the section 3.3 and 4.3 in the revised manuscript.

Line 280: Was Chla measured at beginning of experiments? This could better support statements about biomass productivity altering rate measurements (Cho and Azam 1990).

We are sorry for not measuring Chl *a* at beginning of experiments. We have the Chl *a* data at these stations. It may not have a big difference from the Chl *a* concentrations at beginning of experiments because the seawater used for experiments were also sampled from the Niskin bottles. We have discussed the relationships between Chl *a* and reaction rates in the revised manuscript, as indicated below.

"In addition, almost all the production/degradation rates during summer and winter were independent with Chl *a*, which were also consistent with the results of Motard-Côté et al. (2016) and Tyssebotn et al. (2017)." (L393-394)

General comment: There is a significant order of magnitude difference between absolute AA concentrations presented here and recently published measurements from the Gulf of Mexico. As well, uptake rates of AA were are an order of magnitude less than the degradation rates of AA measured here (Tyssebotn et al. 2017). Please acknowledge these previous measurements and describe potential reasons for differences. The AA dynamics presented here by Wu et al. are an exciting contribution to our knowledge of AA and should be compared with all previous work.

Thank you for your suggestions. We have compared the absolute AA concentrations and degradation rates of AA with previous work and explored the reasons for the differences in the revised manuscript, as indicated below.

"AAd concentrations in the BS and YS during summer were an order of magnitude higher than those (0.8-2.1 nmol L$^{-1}$, median 1.5 nmol L$^{-1}$) in the northern Gulf of Mexico in September 2011 (Tyssebotn et al., 2017). The reasons for these differences might be related to differences in sample storage, analytical methods and study areas. We stored samples at 4 °C, while Tyssebotn et al. (2017) stored at -20 °C. In addition, our study area was highly affected by anthropogenic activities." (L286-291)

"The microbial degradation rates of AAd in the BS and YS during summer were extremely higher than the total biological uptake of AAd (0.07-1.8 nmol L$^{-1}$ d$^{-1}$) in the northern Gulf of Mexico in September 2011 (Tyssebotn et al., 2017), which might be due to the differences in the initial concentrations. Specifically, our study added exogenous AAd at the beginning of incubation. Nevertheless, we both found the microbial degradation rates at inshore stations were higher than those at offshore stations. In addition, almost all the production/degradation rates during summer and winter

were independent with Chl *a*, which were also consistent with the results of Motard-Côté et al. (2016) and Tyssebotn et al. (2017)." (L389-394)

General comment: I recommend the authors consider how the measurements of AA dynamics here could help inform a better understanding of the bacterial 'switch' hypothesis for which the environmental drivers of are still debated (Kiene et al. 2000; Slezak et al. 2007; Levine et al. 2012).

Thanks for your suggestions. We have discussed how bacteria species and abundance affect the microbial degradation of AA in the revised manuscript, as indicated below.

"In addition, the seasonal differences of bacteria abundance and light intensity also made great contributions to the different rates of microbial degradation and photochemical degradation, respectively. According to Liang et al. (2019), the abundances of *Vibrio* (γ-proteobacteria) averaged $1.4 \times 10^6$ copies $L^{-1}$ in summer, which is significantly higher than in winter (Mann-Whitney test, $p < 0.01$), with a mean value of $1.9 \times 10^5$ copies $L^{-1}$. Significant seasonal differences in total bacterial abundance were also observed (Mann-Whitney test, $p < 0.001$). Meanwhile, the average light intensity in summer was 49400 lx, which was also higher than that in winter (34050 lx). All those factors led to high degradation/production rates in summer. In addition, Liang et al. (2019) also found that the dominant bacteria groups displayed different changing patterns in their abundance with seasons and sea areas. Specifically, the abundance of *V. campbellii* was higher in the YS than in the BS in summer ($p < 0.05$), whereas the abundance of *V. caribbeanicus* drastically decreased from the BS to the YS ($p < 0.05$). Therefore, the different microbial degradation/production rates of DMSPd, DMS, and AAd in different sea areas might result from the differences in bacteria species and abundance in the BS and YS. Moreover, the capabilities of diverse bacteria species to degrade AAd were different, which resulted in the inconsistence of AAd microbial degradation rates and rate constants in the comparison between inshore and offshore stations." (L407-420)

Minor comments
Overall the manuscript should be 'cleaned up' in terms of English but also small text errors. Some errors 'overstate' the significance of the statement, but most do not inhibit reading.

Thank you for your suggestions. We have polished the entire manuscript and corrected text errors and wording errors.

Line 29: Please rephrase statement about acid rain. DMS is correlated with the natural acidity of rain (as stated now, implies that DMS is the cause of acid rain).

Thanks for your suggestion. We have revised this sentence as "DMS is correlated with the natural acidity of rain." (L29)

Line 41: Please rephrase minor producers to 'low producers'.

Thanks for your suggestion. We have replaced 'minor producers' with 'low producers'. (L41)

Line 41: I suggest removing the "For example" part of this sentence as you state all of the well-known high producers (i.e. it is not an example). When describing low producers mention other low producer types (McParland and Levine 2019).

According to the reviewer's suggestion, we have removed the sentence of high producers and describe other low producer types, as indicated below.

"DMSP distributions are also controlled by phytoplankton species, among which diatoms, flagellates, prochlorophytes and cyanobacteria are low minor producers of DMSP (McParland and Levine 2019)." (L40-41)

Line 43: this sentence is repeating line 39, please be more concise and edit accordingly

According to the reviewer's suggestion, we have deleted the sentence in line 43 and rephrased the sentence in line 39, as indicated below.

"As an antioxidant, a cryoprotectant, or an osmolyte in marine phytoplankton, the production of DMSP is influenced by environmental parameters such as salinity (Stefels, 2000), temperature (Kirst et al., 1991), and oxidative stress (Sunda et al., 2002)." (L37-39)

Line 47: AA should be defined here (even though it is properly defined in Abstract)

Yes. We defined AA as the abbreviation of acrylic acid in L44 when it was first mentioned in text.

Line 54: Kinsey and Kieber 2016 is incorrect citation for this statement

We have cited another reference of Noordkamp et al., 2000 for this statement. (L52)

Line 55: The use of 'always' here is too strong for the current state of knowledge

Thank you for your suggestion, we have removed 'always' in the revised manuscript.

Line 58: Alcolombri et al. 2015 is incorrect citation, this paper does not measure anything in situ. Additionally, I would expand these citations as there are so many more studies that have conducted the work described in this sentence besides the two.

We have removed that citation and added others including "Lana et al., 2011; Levine et al., 2012; Tyssebotn et al., 2017". (L56-57)

Line 80: complicate**d**

We have replaced 'complicate' with 'complicated'. (L82)

Line 86: How was surface sediment sampled? And where? What time of day collected?

Sediments were collected using a stainless-steel box-corer and sub-sampled to a depth of ca. 3 cm. They were sampled at 12 stations shown in Table 1 during summer cruise. We have added the sampling method of surface sediment in the revised manuscript and sampling time in revised Table 1, as indicated below.

"Sediments were collected using a stainless-steel box-corer and sub-sampled to a depth of ca. 3 cm at 12 stations shown in Table 1 during summer cruise." (L96-97)

Table 1 The AAd concentrations in porewater of surface sediments and in bottom seawater during summer 2015.

| Station | H10 | H12 | H14 | H16 | H19 | H23 | H25 | H26 | B12 | B13 | B61 | B63 |
|---|---|---|---|---|---|---|---|---|---|---|---|---|
| Sampling time | 08-19 | 08-19 | 08-19 | 08-20 | 08-20 | 08-21 | 08-21 | 08-21 | 08-28 | 08-28 | 09-02 | 09-02 |
| | 06:59 | 15:28 | 21:48 | 03:11 | 14:35 | 00:21 | 08:03 | 11:24 | 17:20 | 19:58 | 14:42 | 19:54 |
| Porewater AAd ($\mu$mol $L^{-1}$) | 34.54 | 13.52 | 99.89 | 38.36 | 128.61 | 136.42 | 99.45 | 122.68 | 41.31 | 46.50 | 15.63 | 102.40 |
| Bottom AAd (nmol $L^{-1}$) | 14.34 | 13.41 | 12.32 | 17.54 | 15.59 | 13.25 | 16.23 | 19.01 | 16.74 | 102.98 | 18.95 | 23.68 |

Line 91: How was DMS sampled?

Water samples were transferred from the Niskin bottles to 250 mL brown glass bottle through silicone tubing. While filling the bottles, the samples were allowed to overflow from the top of the bottle to eliminate any headspace in an effort to minimize partitioning into the gas phase. A 2 mL aliquot of seawater sample extracted from the 250 mL brown glass bottle using a 2 mL glass syringe and filtered by syringe filtration through 25 mm Whatman glass fiber (GF/F) filter (Li et al., 2016) was directly injected into a glass bubbling chamber. We have added these sentences in the revised manuscript. (L93-96, L100-102)

Line 94: Was the pre-filtered DMS sample gravity filtered? Please provide a citation for this method. Also, what size GF/F filter was used?

The pre-filtered DMS sample were filtered by syringe filtration through 25 mm GF/F filter. We have added a citation for this method. (L101-102)

Line 101, 117, 120: Were analytical samples run? (in duplicate, triplicate?)

Analytical samples were run in duplicate. We have added this sentence at the end of Section 2.3 in the revised manuscript. (L135-136)

Line 102: Again, what size GF/F filter was used?

47 mm GF/F filter was used here. We have added the size in the revised manuscript. (L109)

Line 104: How long were the samples allowed to oxidize for?

The samples were allowed to oxidize for 2 days. We have added a sentence in the revised manuscript as indicated below.
    "To fully oxidize pre-existing gaseous DMS, the DMSPt and DMSPd samples were incubated in the dark at room temperature for 2 days." (L112-113)

Line 124: Has this methodology for incubations been performed before? Please cite if so.

Yes. GBT inhibition method for DMSPd degradation was performed by Kiene and Gerard (1995). Methods for photochemical and microbial degradation of AA were performed by Wu et al. (2015).

We have added these citations in the revised manuscript. (L145, L157, L163-164)

Line 126: Why were syringes used for incubation? Were they gas-tight?

Yes. These syringes were gas-tight. It was convenient to collect samples at 0, 3, and 6 h if using syringes. We could just push the plunger to let seawater flow out. Meanwhile, it kept the rest seawater in syringes isolated from air.

Line 131: I don't believe Kiene et al. or Vila-Costa et al. address preferential GB uptake?

We are sorry for misunderstanding these articles. We have revised that sentence and cited another reference, as indicated below.
      "and acts as a competitive inhibitor of DMSP (Kiene et al., 1998)." (L146-147)

Line 132: Please address how rates were calculated? Were regressions/fits statistically significant?

According to the reviewer's suggestion, we have added description about rates calculation, as indicated below.
      "Linear regression equations were fit to the DMSPd, DMS and AAd time course data and the apparent rates were estimated as the differences between the slopes of samples with and without GBT." Regressions at most stations were statistically significant. (L148-149, L155-157, L162-164)

Line 147: Kiene 1996 is incorrect citation, they did not measure AA in their study?

No, Kiene (1996) did not measure AA in his study. He determined the kinetics of DMSP(d) degradation by running one with spike additions of DMSP and the other one without additions as control. We thought this method could be applied to AA degradation, so we cited this article. We have removed this citation as the reviewer suggested.

Line 150: Suggestion if you do split into a Results and Discussion section… results for Section 3.1 and 3.2 should be combined for a clearer description of the differences in summer and winter.

According to the reviewer's suggestions, we have split into Results and Discussion sections and combined results for Section 3.1 and 3.2.

Line 151: How are the contours spaced? Center of sea contour looks like 5 ug/L, not 7.07ug/L?

Kriging method was used for interpolating contours. These circles inside the contour of 5 $\mu$g L$^{-1}$ were too small to be marked as their real concentrations. As we measured, the center point was station B61 with the Chl $a$ concentration of 7.07 $\mu$g L$^{-1}$.

Line 163, 170: Chengshan Cape and Changjiang Estuary not on maps

According to the reviewer's suggestion, we have added Chengshan Cape and Changjiang Estuary on maps.

Line 172: The comment about MS thesis belongs in methods

According to the reviewer's suggestion, we have moved the comment about MS thesis to Material and Methods section. (L117-118)

Line 178-192: I suggest moving results for porewaters to be part of the depth profile results as it seems more relevant to depth distributions, not surface distributions.

According to the reviewer's suggestion, we have moved results for porewaters to be part of the depth profile results. (L234-240, L368-387)

Line 185-187: This sentence should be re-written for clarity

According to the reviewer's suggestion, we have re-written this sentence, as indicated below.
   "Wang (2015) reported δ- and γ-proteobacteria were the dominant taxa in the sediments of the BS and the YS, proportion ranging between 24%-70%. Meanwhile, DddY, which is the only known periplasmic DMSP lyase (Li et al., 2017), is widely present in δ- and γ-proteobacteria and can cleave the large amounts of intracellular DMSP (mmol $L^{-1}$ levels) concentrated by DMSP catabolizing bacteria (Wang et al., 2017)." (L375-378)

Line 198: 'was not correlated with' (remove the word 'any')

According to the reviewer's suggestion, we have removed the word 'any'.

Line 203: I think this should read 'main phytoplankton type'? Species likely changed based on the small/big cell statement following

Yes, we have revised 'species' to 'types'. (L309)

Line 204: should read 'small diatoms in winter and larger diatoms in summer'

According to the reviewer's suggestion, we have revised to 'small diatoms in winter and larger diatoms in summer'. (L310)

Line 214: As you discuss everything in context of North to South in the preceding text, for clarity I would order these transects in the same way (same for the order in Figures 4,5 and Table 1).

Thanks for your suggestion, we have ordered these transects North to South in text, Figures 4, 5 and Table 1.

Line 213: If you split Results section, results of Section 3.3 and 3.4 could be combined for clarity.

According to the reviewer's suggestion, we have split into Results and Discussion sections and combined results of Section 3.3 and 3.4 in the revised manuscript.

Line 218: "Concentrations of Chla, AA, DMS" remove this sentence, it should be a part of caption.

According to the reviewer's suggestion, we have removed that sentence in the revised manuscript.

Line 219: 'Both Chla and DMS did not displayed…' this sentence does not make sense to me

We have revised that sentence to 'but Chl *a* and DMS did not displayed…'. (L213)

Line 230: suggested change "…and highest concentrations were observed in…"

Thank you for your suggestion. We have changed that sentence to "…and highest concentrations were observed in…". (L201)

Line 239: Correlations are not causation… using the word 'prove' is an overstatement, please edit.

Thank you for your suggestion. We have changed the word 'prove' to 'indicate'. (L340)

Line 241: I'm confused by this statement. What did Asher et al. 2017 find that indicates the order of average concentrations demonstrates that both DMSPd and DMSPp produce DMS? Please edit.

Asher et al. (2017) referred different sources of DMS including the intra-cellular cleavage of phytoplankton DMSPp catalyzed by the enzyme DMSP lyase and the photochemical and biological reduction of DMSO to DMS in the Introduction section. Here we thought higher of DMS than DMSPd meant DMS is not merely produced through the cleavage of DMSPd, so we cited Asher's paper. We have revised this sentence, as indicated below.
    "The higher values of DMS than DMSPd might be produced through the intra-cellular cleavage of phytoplankton DMSPp catalyzed by the enzyme DMSP lyase and the photochemical and biological reduction of dimethylsulfoxide (DMSO) to DMS (Asher et al., 2017)." (L342-344)

Line 255-260: DMS(P) correlations with both salinity and temperature may be due to a cocorrelation of these abiotic parameters themselves, please use caution in stating these conclusions and incorporate statistical tests appropriately.

We agree with the reviewer. We have added this comment in the revised manuscript, as indicated below.
    "DMSP showed positive correlations with temperature and negative correlations with salinity along the three transects during summer, while DMS and DMSP presented negative correlations with temperature and salinity during winter, which might be due to a co-correlation of these abiotic parameters themselves." (L333-336)

Line 258-259: Kiene and Service 1991 looked at DMS production from *dissolved* DMSP, not particulate. I believe you are discussing a correlation of *total DMSP*. Please edit for clarity.

Thanks for your suggestion. We have deleted this sentence in the revised manuscript.

Line 269: This should be "Figs. 4 and 5" I believe?

We are sorry for the typos. We have corrected "Figs. 3 and 4" to "Figs. 4 and 5" in the revised manuscript. (L360)

Line 294: should be "low bacterial abundance" instead of 'poor'

We have deleted this sentence in the revised manuscript.

Line 336: what does "and so on" refer to? An unknown source? Please be more specific.

According to the reviewer's suggestion, we have revised "and so on" to "other unknown sources". (L429)

Figure/Table comments
Figure 2: I find the labels of summer/winter and Chla/AA on the plots very helpful but please also label the panels (a,b,c,d) in figure and reference in the caption (consistent with other figures)

According to the reviewer's suggestion, we have labeled the panels (a, b, c, d) in Figure 2 and referred them in the caption, as indicated below.

[Figure]

Fig. 2. Horizontal distributions of Chl *a* (μg L⁻¹) and AAd (nmol L⁻¹) in the surface water of the BS and YS during summer and winter. a: Chl *a* in summer; b: AAd in summer; c: Chl *a* in winter; d: AAd in winter.

Figure 3: As mentioned above, I do not think this figure is necessary and it may be more useful to replace with similar surface plots of DMS and DMSP for summer and winter (like Figure 2).

According to the reviewer's suggestion, we have removed that figure and replaced with surface plots of DMS and DMSP for summer and winter shown in Jin (2016) and Sun (2017), as indicated below.

Summer

[Figure]

Winter

[Figure]

Fig. 3. Horizontal distributions of DMS (nmol L$^{-1}$), DMSPd (nmol L$^{-1}$), and DMSPp (nmol L$^{-1}$) in the surface water of the BS and YS during summer and winter. Data in summer and winter presented here were described by Jin (2016) and Sun (2017) respectively.

Figure 4 and 5: Again, reversing the order that the transects are presented to be North to South will make figure clearer. Also the method for interpolating contours should be reported (either in figure captions or methods), and the black dots (I assume sampling points) should be described. Adding temperature for other transects would make for better consistency.

We have ordered the transects North to South, reported the kriging method for interpolating contours in figure captions, described the black dots as sampling points, and added temperature for other transects, as indicated below.

"Fig. 4. Vertical profiles of temperature (°C), Chl *a* (μg L$^{-1}$), AAd (nmol L$^{-1}$), DMS (nmol L$^{-1}$), DMSPd (nmol L$^{-1}$), and DMSPt (nmol L$^{-1}$) along transect B57-63, transect B12-17, and transect H19-26 during summer. Kriging method is used for interpolating contours. The black dots represent sampling points.

Fig. 5. Vertical profiles of temperature (°C), Chl *a* (μg L$^{-1}$), AAd (nmol L$^{-1}$), DMS (nmol L$^{-1}$),

DMSPd (nmol L$^{-1}$), and DMSPt (nmol L$^{-1}$) along transect B12-16 and transect H19-26 during winter. Kriging method is used for interpolating contours. The black dots represent sampling points."

Table 1: Again, I recommend ordering table to be North to South. Caption should better define what 'Surface' refers to (all three sampling sites?) and what depth the transect values reported are.

We have ordered the transects North to South. 'Surface' refers to "surface seawater of the whole study area (the BS and YS)". The transect values are the average of the whole vertical profile of each transect. We have defined these in Table 1 caption in the revised manuscript, as indicated below.

Table 1 Summary of the mean values (ranges) and the significance of seasonal differences of AAd, DMS, DMSPd, and DMSPt at surface seawater of the BS and YS and at whole vertical profiles of transects during summer and winter. The significance of seasonal differences was obtained using Mann-Whitney test.

| | | AAd (nmol L$^{-1}$) | DMS (nmol L$^{-1}$) | DMSPd (nmol L$^{-1}$) | DMSPt (nmol L$^{-1}$) |
|---|---|---|---|---|---|
| Summer | Surface | 30.01 ± 21.12 (10.53-92.29) | 6.12 ± 3.01 (1.10-14.32)* | 6.03 ± 3.45 (1.05-13.23)* | 28.86 ± 14.15 (8.70-63.03)* |
| | B57-63 | 36.36 ± 23.57 (11.08-73.06) | 5.51 ± 2.01 (2.57-8.79) | 1.56 ± 0.84 (0.72-3.37) | 22.94 ± 21.28 (4.12-56.61) |
| | B12-17 | 34.60 ± 26.00 (12.77-102.98) | 7.37 ± 4.50 (0.74-15.76) | 1.12 ± 0.48 (0.36-2.01) | 15.45 ± 17.98 (1.90-63.03) |
| | H19-26 | 22.24 ± 18.25 (13.19-85.86) | 6.44 ± 5.14 (0.79-21.98) | 3.05 ± 4.92 (0.61-21.59) | 13.67 ± 12.90 (1.11-55.14) |
| Winter | Surface | 14.98 ± 7.22 (4.28-42.05) | 1.38 ± 0.41 (0.54-2.22)* | 2.30 ± 0.80 (1.16-4.29)* | 10.39 ± 4.14 (2.36-22.21)* |
| | B12-16 | 17.68 ± 5.21 (13.94-27.69) | 1.99 ± 1.02 (1.12-4.56) | 2.92 ± 0.82 (1.54-4.55) | 11.44 ± 5.89 (5.33-24.50) |
| | H19-26 | 17.08 ± 6.72 (11.04-39.47) | 0.96 ± 0.29 (0.52-1.35) | 3.06 ± 1.07 (1.92-6.06) | 11.88 ± 3.97 (6.12-19.92) |
| Seasonal difference | Surface | $p < 0.001$ | $p < 0.001$ | $p < 0.01$ | $p < 0.001$ |
| | B12-16 | $p < 0.05$ | $p < 0.05$ | $p < 0.001$ | |
| | H19-26 | | $p < 0.001$ | $p < 0.01$ | |

* collected from published MS theses (Jin, 2016; Sun, 2017)

Table 2: What correlation test was used? Additionally, please address in the methods how temperature and salinity were measured (i.e. CTD profile or was salinity actually measured?).

Pearson correlation test was used here. We have added this sentence in figure caption as indicated below and addressed CTD profiles of temperature and salinity in Material and methods section. (L93)

"Table 2 Correlations between AAd, DMS, DMSP, and other biogeochemical parameters in the BS and YS during summer and winter. Pearson correlation test was used here."

Table 4: Very minor, but the table would be easier to read if the abbreviation of BS and SYS are added above the transect station names. Also, these experiments were reported to be conducted in duplicate so please report biological errors for rate measurements.

Thank you for your suggestions. We have added the abbreviation of BS and SYS above the transect station names and reported standard errors for rate measurements in revised Table 4, as indicated below.

[revised manuscript text omitted]

---

## Author Response (AR2)

Dear Prof. Dai,

Thank you for your kind consideration and constructive comments for our manuscript entitled 'Acrylic acid and related dimethylated sulfur compounds in the Bohai and Yellow Seas during summer and winter'. We are grateful to the anonymous reviewer for his/her constructive suggestions, which is of great help to improve the manuscript. Please find our final responses (in blue) and changes (in red) to all comments (in black) in this document.

**Response to reviewer**

Comments from the reviewer are in black while our responses are in blue and changes in the manuscript are in red.

The revised manuscript is an improvement. The authors took the comments seriously and tried to revise as recommended. However, there are still a few problems (see specific comments below). Also, the English needs to be proofed, as there are still many mistakes.

We are very grateful to the reviewer for all the constructive comments and helpful suggestions to improve this manuscript.

We have carefully considered the reviewer's comments below and conducted the revision seriously. Also, we had our revised manuscript proof-read by a native English speaker.

Specific comments

Response, page 8; Manuscript, lines 206, 292, and Table 2 - I am a bit confused about the temperature and salinity correlations. The other reviewer also expressed concern here and the revision does not seem to answer the question. If AAd has a terrestrial (riverine source) why is there no anticorrelation between salinity and AA. Why is the anticorrelation always with DMS/P?

According to the other reviewer's comments, DMS(P) correlations with both salinity and temperature may be simply due to a co-correlation of these abiotic parameters (temperature and salinity) themselves. Temperature decreases and salinity increases with depth generally in summer and they do not change apparently with depth in winter due to the vertical mixing, thus DMSP showed opposite correlations with temperature (positive) and salinity (negative) during summer and same correlations with both temperature and salinity (negative) during winter along the three transects. There is no anticorrelation between salinity and DMS(P) in surface seawater. DMS(P) and AAd distributions were affected by various factors, so they might not present good anticorrelations with salinity. Nevertheless, relatively high AA concentrations in the outer Yellow River and Yalu River estuaries and around densely populated Chengshan Cape still could reveal the terrestrial sources of AAd. We will conduct investigations in the inner estuaries and at sewage outfalls to further demonstrate the terrestrial sources of AAd in the future work.

Line 41 – I think is not clear why the authors only talk about low producers of DMSP. They should give examples of both high and low producers.

Thank you for your suggestion, we have added examples of high producers in the revised manuscript as below.

    "…, among which coccolithophorids, dinoflagellates, and prymnesiophytes are the high-producing algae of DMSP (Keller et al., 1989), and diatoms, flagellates, Prochlorophytes and cyanobacteria are low producers of DMSP (McParland and Levine, 2019)." (L43-45)

Lines 58-61 – I assume this is not a comprehensive compilation of AA studies, correct? Why are only these presented? I assume it is because of the region. However, the authors state that the biogeochemistry of AA has received little attention, which I assume is more global in scope. Are there other studies that should be referenced here?

Thank you for your suggestion, we have referenced other studies about the biogeochemistry of AAd in the revised manuscript as below.
"Recently, the biogeochemistry of AA in the oceans and the roles of AA in the marine sulfur cycle and the microbial community have received increasing attention globally. Kinsey et al. (2016) explored the effects of iron limitation and UV radiation on Phaeocystis antarctica growth and AA concentrations. The concentrations, biological uptake, and respiration of dissolved AA (AAd) were investigated in the northern Gulf of Mexico (Tyssebotn et al., 2017)." (L61-64)

Line 255 - What is meant by rational and what is the meaning of the whole sentence?

We have revised this sentence as "Although the rates of AAd microbial degradation at all stations were extremely high compared to the rates of AAd production and AAd photochemical degradation due to the addition of exogenous AAd at the beginning of incubation, the measured rates still reflect the capability of bacterially mediated degradation of AAd." (L270-273)

Line 258 - Why say assuming first order when the authors measured this themselves? It is not necessary to assume, but it should be stated that there is evidence for first order kinetics.

Thank you for your suggestion, we have revised this sentence as "Since the DMSPd and AAd degradation follow first-order kinetics (Kiene and Linn, 2000a; Wu et al., 2015), the turnover times of DMSPd and the rate constants of the AAd microbial and photochemical degradation were calculated (Table 4)" and added two references to provide evidence for the first order kinetics. (L275-277)

Lines 287-291 - Did anyone test these storage effects?

We are sorry for not finding articles about testing the storage effects, but we did storage experiments in lab when we developed the HPLC method for AAd determination. We found the concentrations of AAd would not change in 25 days when storing samples at 4 °C. However, we did not compare this storage method with storing samples at -20 °C. We will improve this method in future work.

Lines 332-340 - What do the correlations presented in this paragraph mean? There is no real discussion of these.

We agree with the reviewer. There is little discussion in this paragraph, therefore, we have moved it to Results section. (L241-249)

Line 424 – Is it reasonable to assume steady state here? Please state why.

It is reasonable to assume steady state here. We applied a simple box model here which is based on

steady state, namely, the input fluxes balance the output fluxes. Furthermore, the AAd concentrations would not fluctuate drastically during our study periods (20 days in summer and 19 days in winter). Therefore, $dc/dt = 0$ is reasonable.

Lines 431-432 – What is meant by coincided and the whole sentence?

We have revised this sentence as "The relationship of the rates from other sources between summer and winter was similar to that of the AAd concentrations in the surface seawater between summer and winter; namely, the rate from the other sources and the AAd concentrations in the surface seawater in winter were less than half of those in summer.". (L444-447)

Line 443 – Why should anthropogenic sources of AAd decrease in winter? Or is the decrease from summer to winter dominated by the natural source?

Because rivers' discharge fluxes in winter are lower than those in summer, they bring less anthropogenic AAd to the study area. Yes, natural sources like production from DMSP also play important roles in decreasing AAd concentrations in winter because decreased temperature and phytoplankton amounts and other factors may weaken the degradation of DMSP. In a word, low AAd concentrations in winter were the combined result of decreasing natural sources and anthropogenic sources of AAd. We have explained this question in Section 4.2.

[revised manuscript text omitted]